# Applications of Wireless Sensor Networks and Internet of Things Frameworks in the Industry Revolution 4.0: A Systematic Literature Review

**DOI:** 10.3390/s22062087

**Published:** 2022-03-08

**Authors:** Mamoona Majid, Shaista Habib, Abdul Rehman Javed, Muhammad Rizwan, Gautam Srivastava, Thippa Reddy Gadekallu, Jerry Chun-Wei Lin

**Affiliations:** 1School of System and Technology, University of Management and Technology, Lahore 54782, Pakistan; mamoona.majid@umt.edu.pk (M.M.); shaista.habib@umt.edu.pk (S.H.); 2Department of Cyber Security, PAF Complex, E-9, Air University, Islamabad 44000, Pakistan; abdulrehman.cs@au.edu.pk; 3Department of Computer Science, Kinnaird College for Women, Lahore 54000, Pakistan; muhammad.rizwan@kinnaird.edu.pk; 4Department of Mathematics and Computer Science, Brandon University, Brandon, MB R7A 6A9, Canada; srivastavag@brandonu.ca; 5Research Center for Interneural Computing, China Medical University, Taichung 406040, Taiwan; 6School of Information Technology and Engineering, Vellore Institute of Technology, Vellore 632014, Tamil Nadu, India; thippareddy.g@vit.ac.in; 7Department of Computer Science, Electrical Engineering and Mathematical Sciences, Western Norway University of Applied Sciences, 5063 Bergen, Norway

**Keywords:** Internet of Things (IoT), industrial revolution 4.0 (IR 4.0), computer networks, network security, wireless sensor networks (WSN), systematic literature review (SLR), state-of-the-art

## Abstract

The 21st century has seen rapid changes in technology, industry, and social patterns. Most industries have moved towards automation, and human intervention has decreased, which has led to a revolution in industries, named the fourth industrial revolution (Industry 4.0). Industry 4.0 or the fourth industrial revolution (IR 4.0) relies heavily on the Internet of Things (IoT) and wireless sensor networks (WSN). IoT and WSN are used in various control systems, including environmental monitoring, home automation, and chemical/biological attack detection. IoT devices and applications are used to process extracted data from WSN devices and transmit them to remote locations. This systematic literature review offers a wide range of information on Industry 4.0, finds research gaps, and recommends future directions. Seven research questions are addressed in this article: (i) What are the contributions of WSN in IR 4.0? (ii) What are the contributions of IoT in IR 4.0? (iii) What are the types of WSN coverage areas for IR 4.0? (iv) What are the major types of network intruders in WSN and IoT systems? (v) What are the prominent network security attacks in WSN and IoT? (vi) What are the significant issues in IoT and WSN frameworks? and (vii) What are the limitations and research gaps in the existing work? This study mainly focuses on research solutions and new techniques to automate Industry 4.0. In this research, we analyzed over 130 articles from 2014 until 2021. This paper covers several aspects of Industry 4.0, from the designing phase to security needs, from the deployment stage to the classification of the network, the difficulties, challenges, and future directions.

## 1. Introduction

Smart technologies play a crucial role in sustainable economic growth. They transform houses, offices, factories, and even cities into autonomic, self-controlled systems without human intervention [1]. This modern automation trend and ever-increasing use of cutting-edge technologies are boosting the world’s economy [2]. The Internet of Things (IoT) and Wireless Sensor Networks (WSN) both play vital roles in this modernization [3]. IoT is a branch of engineering primarily concerned with offering thousands of miniature, physical connected objects, which may collaborate to achieve a shared goal. IoT has gained much importance due to the abundant usage of these tiny networked devices. These are smart, yet basic things that can sense and communicate wirelessly [4]. WSN is a collection of sensor and routing nodes, as shown in Figure 1, which may be put together in the environment to predict physical conditions, such as wind, temperature, and many others. These networks collect and process data from tiny nodes and then transfer it to the operators. Figure 2 illustrates that sensor networks are used in a variety of control systems, including environmental monitoring, home automation, chemical and biological assault detection, smart grid deployment [5], surveillance, and many more. WSN also plays a significant role in aquaculture and the oil industry, including data collection, offshore exploration, disaster prevention, tactical surveillance, and pollution monitoring [6,7,8].

WSN are often deployed in remote areas where human intervention is not possible for post-deployment maintenance. Therefore, efforts are being made to enhance their efficiency and durability [9]. There are many barriers to WSN deployment, such as power consumption–long-distance deployment. Due to automation trends and applications developed, these barriers are no longer barriers for large-scale remote deployment. In general, WSN follows a star topology to decrease the network failure probability by connecting all systems to a central node. While ad-hoc networks follow mesh topology where each node is human-driven [10].

In physical production systems, grid and energy-saving applications minimize the energy resources and noise pollution. In the last few decades, transportation has improved a lot with the usage of smart IoT devices, such as signals and high-resolution cameras on roads, which has led to an increase in traffic flow. RFID readers are deployed at toll booths that automatically deduct toll amounts after reading RFID tags on vehicles. In the transportation sector, smart vehicles reduce the travelling time and also fuel consumption with low cost of mobility and reduced human efforts [11], atmospheric monitoring reduces pollution, and surveillance applications reduce crime. Nowadays, WSN also plays a role in precision agriculture. On the other hand, WSN applications facilitate our day to day lives, making them more comfortable, such as healthcare applications that improve our health and longevity.

Besides WSN, IoT has also played an important role in human life. IoT and the digital age play essential roles in overcoming social and physical barriers and providing ease and mobility to people, resulting in improved and equal opportunities, and access to information [12,13]. IoT also has many application areas such as agribusiness, climate, clinical care, education, transportation, and finance, as shown in Figure 2.

In regard to information and communication technology, researchers are attracted to IoT [14]. By adopting this essential technology, companies have become smarter, more competitive, automated, and sustainable in the global supply chain. In today’s competitive marketplace, supply chains are struggling as they compete with each other. Therefore, IoT devices are an effective way to authenticate, monitor, and track products using GPS and many technologies [15,16]. Industry 4.0 stands for the fourth industrial revolution in the digital age, it is associated with virtualizing real-world scenarios of production and processing without human intervention. This virtual world is linked to IoT devices, allowing the creation of cyber–physical systems to communicate and cooperate [17,18]. This fully connected manufacturing system—operating without human intervention by generating, transferring, receiving, and processing necessary data to conduct all required tasks for producing all kinds of goods—is one of Industry 4.0’s key “constructs”. The concept of Industry 4.0 is based on the combination of three main elements: people, things, and business [19]. A complete cyber–physical production system created by the integration of IoT devices, things and objects (IoT), sensor nodes (WSN), and people, is shown in Figure 3. CPS is a typical example of Industry 4.0. IoT is the connection of smart devices, objects, or machines to the internet and with each other. In WSN systems, there is no direct connection of these devices to the Internet. These systems can send their data to the Internet by connecting several sensor nodes to a central routing node. While CPS systems involve the integration of IoT devices, computation, networking, and physical process, IoT is an essential component of CPS. CPS systems are key elements in the implementation of IR 4.0 [20]. Industry 4.0 is the network-enabled entity that automates the whole process of manufacturing, connecting business and processes. Market demands and the advancements in new technologies are transforming manufacturing firms’ business operations into smart factories and warehouses. Due to this automation, IoT devices are producing a massive amount of data daily, known as big data [21,22]. Statistics show that, at the end of 2021, there were more than 10 billion active IoT devices globally [23]. By 2030, the number of active IoT devices is expected to exceed 10 billion to 25.4 billion. By 2025, the data created by IoT devices will reach 73.1 ZB (zeta bytes) [24]. In 2020, the IoT industry was predicted to generate more than USD 450 billion, including hardware, software, systems integration, and data services. By the end of 2021, it reached USD 520 billion. The global amount expected to be spent on the IoT in 2022 is USD 1 trillion. The IoT industry is predicted to grow to more than USD 2 trillion by 2027 [25,26]. The increasing number of devices and the usage by humans shows the importance of IoT devices; moreover, the industry is growing and gaining revenue.

In this paper, we conduct a detailed systematic literature review on the applications and contributions of IoT and WSN in Industry 4.0. We had a large corpus of data to analyze papers using systematic approaches. Among the selected articles, we obtained 22 articles describing the detailed review of existing security techniques, applications used, advantages/disadvantages, and limitations of IoT and WSN. Most of them reviewed the articles in terms of application of IoT and attacks only. The paper mainly focuses on research challenges, issues, limitations, and the future direction of IoT and WSN frameworks in Industry 4.0.

### 1.1. Motivation

The primary motivation for performing SLR is the ever-increasing trend of automation in Industry 4.0. Industry 4.0 is made up of both WSN technology and IoT to digitize work. Over time, we see how digitization and technology are replacing people in the workplace, and dramatically changing the global workforce. Technology has brought ease to human life and the efficiency of things by making them eco-friendly, more agile, and productive. From smart cities to smart industries, a drastic change has happened due to the intelligent usage of IoT and WSN and IR 4.0. For example, a smart factory integrates virtual and physical systems and calibrating instruments to record their readings immediately. In short, the integration of IoT and WSN with Industry 4.0 has reduced labour needs, freed humans from low-level skilled work, and improved the quantity and quality of work. Therefore, To achieve better analysis results, we systematically analyzed almost all research data related to IoT and WSN domains in Industry 4.0.

### 1.2. Contribution

From smart factories to individual lives, IoT and WSN play significant roles. There are many advantages, but security problems have arisen as these devices generate considerable amounts of data daily. These papers amalgamated both sides: IoT and WSN contributions and their security risks. This paper focuses more on the contributions of IoT and WSN in Industry 4.0 and presents an in-depth review and analysis of IoT and WSN. The paper’s main contribution involves formulating research questions from filtered data and building a strong work foundation using SLR methods. We discuss various applications and security attacks in IR 4.0. Furthermore, new findings of the paper pertain to the key challenges and open issues of WSN integrated with IoT in Industry 4.0, to optimize different criteria.

### 1.3. Paper Organization

The remainder of the paper is organized into the following sections. Table 1 presents the notation used in the entire paper. Section 2 provides the related works. Comparative analyses of these review papers are given in Table 2. Section 3 presents a detailed systematic literature survey (SLR). Research questions presented in Table 3 are addressed in Section 4. Section 5 presents the challenges and issues. Future directions are elaborated in Section 6. Finally, the conclusion is presented in Section 7.

## 2. Related Studies

In this section, we analyze the state-of-the-art research studies on IoT and WSN. With the fourth industrial revolution, it is observed that communication, computation, and storage costs have remarkably decreased, which make integration of IoT and WSN possible and cost-effective globally. We studied many review articles and original research. Existing review articles lack, in many aspects, research challenges, issues, limitations, and future directions of IoT and WSN, but the systematic literature review (SLR) given in this work is precise enough to deal with the aspects of IoT and WSN area. Related work of review papers is provided next. Moreover, Table 2 is presented which shows the comparison between the proposed research work and the existing state-of-the-art analysis.

The authors in [1] discussed the very novel technology “WiFi”. They discussed how this technology helps IoT devices–used in various applications, such as smart cities, healthcare systems, and smart houses—communicate effectively. Landaluce et al. [4] discussed RFID and WSN technology in detail. They discussed how RFIDs are used to trace devices while WSN gathers information about them from interconnected devices. The authors also discussed the obstacles and challenges, such as energy consumption, fault-tolerant, communication interference, and cost feasibility, along with detailed surveys. They provided the advantages and limitations of wearable sensor devices. Energy consumption is increasing each day. Therefore, Dailipi [4] explored how IoT evolution has managed the electricity consumption process and provided many benefits to grid stations, consumers, and suppliers. They also discussed the security issues and challenges after introducing IoT devices in smart grids.

In [6], the authors discussed how WSN advancement had played a significant role in UASN. They traced the location of sensor nodes deployed underwater in the ocean using localization algorithms. They reviewed many applications of UASN, their advantages and their disadvantages. They also discussed the challenges they faced during deployment and presented future directions in the acoustic area. In [10], researchers discussed the applications and advantages of WSN being deployed everywhere due to their flexible and dynamic nature. They discussed each application of WSN in urban areas and their solutions. They analyzed how WSN deployment in urban areas demanded much more care and attention due to harsh weather and perverse channel conditions.

WSN is applicable in many domains, such as industrial automation, and the requirement elicitation of the industrial process is different from general WSN requirement gatherings. In [17], the authors presented some standard protocols that were used to measure the requirements of industrial applications. They also provide solutions to WSN protocols by discussing MAC, routing, and transport in detail. They also discussed the security issues in detail and identified the unsolved challenges encountered during designing standard protocols. In [19], the authors conducted SLR, which is mainly focused on scope definition, concept, literature review, analysis, synthesis, and future research directions. Their selected study has contributed to eight thematic perspectives: intelligence factories, CPS, data handling, IT infrastructure, digital transformation, HCI, IoT, and cloud [18].

Due to the rapid evolution in IoT and WSN, technology is becoming more vulnerable to security threats [42,43,44]. Therefore authors in [27] presented threat models for the security of WSN and IoT devices communication. In [28], the authors discussed IoT applications, advantages, challenges, and security issues from both technological and social perspectives. Researchers have provided detailed architectures of IoT and WSN and discussions of IDS system protocols. They also discussed the security challenges and attacks on IoT and WSN communication devices. Moreover, in [29], the authors conducted extensive research related to smart homes, applications, and IoT. They collected 229 articles, analyzed them thoroughly, and divided them into four categories. They discussed smart home IoT applications in the first category. The second category concerned with IoT applications in smart home technology. In the third category, they developed a framework to operate further. In the fourth category, they developed smart IoT home applications.

IoT has dramatically changed human life, especially regarding communication devices integrating technologies. Traditional industry is changing in the digital industry, and WSN and wireless sensor and actuator networks (WSANs) are the core parts of Industry 4.0. In the article [30], the authors discussed the industrial wireless sensor network (IWSN) and industrial wireless sensor and actuator network (IWSAN) in detail. They discussed IWSAN requirements, applications, challenges, solutions, and future directions in detail. IWSN/IWSAN are compelling technologies due to their promising benefits, such as low-cost deployment, less complexity, and mobility support.

In [31], the authors discussed how IoT plays a vital role in bringing the physical world close to the digital world. They discussed technologies, various challenges, future directions, and various Internet of Things (IoT) applications.

In [32], Sharma et al. have described the sensor nodes according to coverage point of view. They analyzed the full coverage issues by considering node type, deployment type, communication and sensing range, and positioning-based independent algorithms. They also discussed the research challenges of WSN.

Andrey et al. [33] described a detailed survey on IDS systems and presented the methods proposed for IoT. Using a cross-platform distributed approach, they analyzed the IDS system, their platform differences, and current research trends in IDS. In [34], the authors analyzed and discussed the solutions to identify and detect sinkhole attacks in the WSN domain. They discussed the advantages and limitations of the proposed solution as well. In [35], the authors presented a detailed review on security attacks of WSN and IoT along with their preventive measures, mitigations, and detection mechanisms. They stated that the integration of IoT and WSN has raised new challenges and open security issues. Although technology has increased, it has become prone to external attacks.

In [36], the authors presented a review on the security of mobile networks. They discussed the integration of WSN with IoT via the Internet and how the inter-connected devices have guarded networks against external attacks, keeping the router in a secure and protected environment. They discussed the attacks and their detection mechanisms over the Internet. Similarly, the authors in [37] have discussed the wormhole attack and its solution in IoT and WSN domains. They stated that the detection algorithm performed much better for IoT (70%) than WSN (20%). In [45], the authors discussed side-channel attacks in smartphones. Similarly, the authors in [38] discussed the security threats, challenges, and solutions in the IoT domain. While in [39], the authors analyzed existing protocols for secure communication between IoT devices. They also discussed open issues and challenges raised during the communication of IoT devices and future directions in IoT. The authors in [40] presented a detailed review regarding deployment schemes, classification, working, and comparative analyses of sensor nodes. This growing technology trend has converged the “world sense” from traditional systems to CPS—this transition is called Industry 4.0. The authors in [41] conducted a bibliometric review of 12 different approaches of critical aspects of Industry 4.0.

From the above-detailed literature review, we noticed that authors and researchers have worked on IoT and WSN, but the integration of both IoT & WSN with Industry 4.0 is benign. They discussed their applications, security attacks, advantages, and limitations at each level.

The proposed paper is more oriented towards the applications and contributions of IoT and WSN in Industry 4.0, along with the security attacks, their challenges, and open issues in each domain. This paper also provides the limitations and future directions for IoT and WSN in Industry 4.0.

## 3. Research Methodology

The SLR followed in this work is based on the template of IEEE SLR; all research steps and guidelines were followed using the same template. The research process and results were manually evaluated by comparing different research methods and techniques presented by different authors based on efficiency, security, limitations, and performance. First, we formulated our research questions based on the reviewed articles and then searched for these keywords in various databases. Using avoidance and consideration criteria, only relevant papers were considered. Then, we evaluated the quality of the papers, and then extracted the relevant features.

### 3.1. Planning Review

The planning of the review was based on the research questions and their objectives. Analyzes of IoT and WSN were conducted to show their importance in daily life. The use of smart and appropriate devices has been studied in this field but still needs critical analysis. Therefore, it is important to make a systematic review article in this realm, especially related to IoT and WSN, to show future directions. Therefore, we established the search technique, search strings, inclusion/exclusion criteria, and quality assessment criteria for the papers collected from different repositories. Figure 4 shows the step of our proposed review planning process.

### 3.2. Research Goals

The main objective of this study is to find out the major contributions, solved problems, and challenges of IoT and WSN in Industry 4.0. In addition, the research gap and limitations of current work in these areas helped us find room for improvement; we also explored future directions and possible outcomes in this area. We designed questionnaires on these domains to find high quality research papers. Some of them are listed in Table 3. In addition, we answer the research questions in Table 3 and in Section 4.

### 3.3. Selection of Primary Studies

It was a challenge to search the specific and limited computer science/engineering databases to get a “complete picture” of the research questions. After formulating the research questions, we collected research papers from various repositories, such as IEEE Xplore, ACM, Wiley Online Search, Elsevier, etc. The papers from electronic databases with the areas of IoT, WSN, and Industry 4.0 were efficiently evaluated. Figure 5 shows the names of the repositories where the research articles were collected from 2014 to June 2021.

### 3.4. Selection/Search Criteria

This SLR combines three domains: IoT, WSN, and Industrial Revolution 4.0, so the search strings were related to each domain and its applications in Industry 4.0. Table 4 shows the subject search strings used to search for relevant articles. The search strings were divided into five groups to search for relevant articles from reputable journals and conferences.

### 3.5. Inclusion and Exclusion Criteria

Several research articles were found on the mentioned domains, IoT, WSN and IR 4.0. To extract the most relevant and concise data from these articles, the following criteria were chosen, as shown in Table 5. The publicly available papers written in English and related to the search strings and research objectives in IoT, WSN, and IR 4.0 published between 2014 and 2021 were considered for further SLR tracking.

### 3.6. Selection Results

As mentioned earlier, the downloaded articles were based on the initial screening processes, which were based on inclusion criteria. The articles were screened using the initial quality assessment criteria (QAC). The main objective of using the QAC was to ensure that the primary studies selected were appropriate to address the concerns of previous studies. Nearly 300 articles were found on the above topics. After applying a duplication filter, 40 articles were discarded, leaving 260 articles that were reviewed based on the above exclusion criteria. In the next step, 60 articles were excluded based on the inclusion/exclusion criteria. A quality analysis was performed for the remaining 200 articles. After the quality analysis, only 120 articles were reviewed. Figure 6 shows the year-by-year distribution of articles reviewed that were used for the SLR study. There is no doubt that the eliminated articles contained valuable material, but they were not studied because they did not meet the screening criteria. The number of articles was initially limited to 120, after which they were stored in the citation manager software for information synthesis.

The selected papers were evaluated against the quality assessment criteria (QAC) listed in Table 6. Table 6 shows the quality assessment criteria (QAC) used to screen the articles for response grading. Studies selected for screening are described in Table 7. Studies with a score greater than or equal to 80 were selected according to the grading criteria shown in Figure 7.

### 3.7. Data Extraction and Synthesis Process

After collecting articles relevant to the research questions and objectives, we performed a SLR according to various characteristics, such as year of publication, limitations, and future work. The information or previously collected characteristics were integrated with the responses collected through questionnaires to summarize the information.

After an extensive and systematic review of the literature, the research questions are answered and described in the following sections. The contributions and types of WSN and IoT are respectively explained in the Section 4.1–Section 4.3. Before attacks network intrusions are briefly classified in Section 4.4. In contrast, network security attacks and WSN and IoT layer issues are discussed in Section 4.5 and Section 4.6. In Section 4.7, we discuss the limitations and future work of the selected papers. Finally, we conclude the paper in Section 7.

## 4. Results

In this section, we have briefly discuss the results of the SLR work. We have formulated the research questions presented in Table 3 and divided the results section into seven subsections to answer them. The information about the contribution of WSN and IoT in IR 4.0, network security attacks and intruders in WSN and IoT, WSN coverage, issues in IoT and WSN framework, and limitations of existing reviews are explained in this section. The challenges section summarizes all the problems encountered in WSN and IoT usage.

### 4.1. RQ1: Contributions of WSN in IR 4.0

The use of WSN has attracted a lot of attention in industry. Because of their prevalence and use in industry, WSN have given rise to IWSN and IWSAN, respectively. These networks enable autonomous work without human intervention. The in-network transmission characteristics are fundamental properties of WSN. Sensor nodes do not transmit raw data, but integrate it to save communication costs. Due to their unique properties and wide range of applications, they are used in many systems, such as military, surveillance, home automation, smart cities, smart buildings, and healthcare monitoring [27]. WSN- and IoT-based devices are used to create reliable, realistic, efficient, flexible, and economical smart cities and buildings in heterogeneous environments [48].

The categories discussed in this paper and the contribution of WSN in IR 4.0 are listed in the form of a taxonomy presented in Figure 8. WSN is also used in health care management systems to monitor medically ill patients, periodically check their various measurements such as blood glucose levels and pulse, and wirelessly transmit this information to a central repository for further diagnosis [49]. WSN is also used to assist elderly and disabled people. Disabled people are informed of relevant information about real-time activities using smart devices, such as wristwatches [28,50]. In recent decades, WSN have been applied in many fields, including transportation, agriculture [51], automation, manufacturing process control, and supply chain management. In addition, WSN can be easily deployed, have low construction cost, no expenditure on wiring, and lower complexity [52,53].

WSN can be used in various manufacturing applications, such as industrial control, process automation, rescue, and defense. WSN is also used to control and automate industrial processes known as actuators. They can operate independently of a physical environment defined by predefined dimensions [54]. WSN is used to collect, track, and record data in smart factories. Data acquisition is usually done by product information in smart factories. After data collection, processing is done by intelligent machines and manufacturing systems. Nowadays, these factories are self-sufficient, cost-effective, and automated by integrating wireless communications with existing private networks and reducing labor [30].

In software, WSN takes maximum advantage of wireless technologies used to build industrial network infrastructure [55]. On the other side, Industry 4.0 is integrating big data analytics and cloud services [56], 3D printing, computer security, autonomous robotics, the Internet of Things (IoT), 5G, Augmented Reality (AR), and modeling [57,58].

### 4.2. RQ2: Contributions of IoT in IR 4.0

An integrated digital system would introduce a new intelligent and economical manufacturing process using cutting-edge technology for a variety of existing items and processes [59]. The data collected from production process warehouses and consumer information can be critically analyzed to make a decision in real time under Industry 4.0. The real-time decision-making capability of each small and medium organization enables them to efficiently accept new technologies [60,61]. Industrial IoT delivers solutions and services that provide insights into an organization’s ability to monitor and control its operations and assets. IIoT software and tools provide important solutions for better process, layout scheduling, organization, and administration.

In addition, IIoT enables real-time and decision-making features among numerous networked devices that can communicate and interact with each other [62]. Because of the rapid communication and data transfer, attackers can attack data and cause harm to an organization, resulting in cyber attacks. Cyber attacks have become a major challenge for the industrial Internet of Things (IIoT). Therefore, integrating IoT with Industry 4.0 plays a critical role in securing IoT devices from attacks. Unique security objectives and challenges of IIoT have been introduced to overcome industrial-level issues. IIoT challenges and objectives relate to IoT being used by consumers and its capabilities leading to longer life of IoT devices and sensor nodes. In [63], the authors analyzed security challenges and attacks at three levels of the network (perception, network, and application). They considered cryptographic challenges, authentication, network monitoring, and access control mechanisms. The IIoT also addresses local network connectivity and protection from attackers inside. Cyber attacks have become a serious challenge for the IIoT. Hackers attack infrastructure/devices through intrusion and hiding, resulting in poor performance. A bidirectional long and short term memory network with a multi-feature layer has been developed to avoid temporal attacks. Machine learning-based networks that learn temporal attacks from historical data and make associations with test data can effectively identify and detect different attacks within different intervals [64].

DL-IIoT has enormous potential to improve data processing and contribute to IR 4.0. Similarly, machine learning algorithms, such as logistic regression, are widely used for malware detection and security threat protection [65]. Deep learning algorithms are also used for intelligent analysis and processing. Deep learning [46] algorithms such as CNN, auto-encoders, and recurrent neural networks have applications such as intelligent assembly and manufacturing, network monitoring, and accident prevention. The application of deep learning algorithms in IIoT has also enabled various smart applications such as manufacturing, active attack detection and prevention systems, smart meters, and smart agriculture [66]. DL-IIoT relies heavily on data collection, which affects the privacy of the organization’s data. Therefore, differentiated privacy is used to protect user privacy, reduce privacy risk, and achieve high performance in IIoT.

On the other hand, IoT and IIoT must provide “differentiated privacy” for individuals and industrial data [67,68,69]. The contribution of IoT in Industry 4.0 has improved the average availability and sustainability of the enterprise by knowing market trends and decreasing unanticipated downturns [70]. The taxonomy of existing studies and the contribution of IoT in IR 4.0 is shown in Figure 8.

### 4.3. RQ3: Type of WSN Coverage Area for IR 4.0

WSN coverage is an important factor in sensor quality. Sensing and connectivity are key features of WSN. The former indicate how well a particular sensor behaves and monitors a particular area of interest in which it is deployed. Connectivity shows how well different nodes communicate with each other. The types of wireless sensor network coverage are as follows.

#### 4.3.1. Area Coverage

Sensors usually perform well in area coverage and monitors the field of interest (FoI). This is also called “blanket coverage” because each node communicates with others. Each sensor is placed so that the coverage of the other WSN sensors covers each other [32].

#### 4.3.2. Barrier Coverage

Barrier coverage of the sensor network comes into play when some intruders try to breach the security layer of the network. Sensors are easy to handle and deploy; therefore, their wireless nature makes them vulnerable to malicious security attacks [71]. The sensor nodes are primarily distributed throughout the network and are deployed in chains to detect interruptions.

#### 4.3.3. Point Coverage

Point coverage aims to find a target within range using nearby nodes. It is also known as target coverage. It is characterized by consuming less energy in a given zone than in the entire region of the FoI. Only a few targets are covered by individual nodes, while other targets can be detected under other sensor scopes. The primary goal is to select a specific target within the FoI to reduce energy consumption [71].

### 4.4. RQ4: Classification of Network Intruders

There are two main types of intruders: internal and external. Internal intruders are people inside the organization; they can be either a customer or a legitimate user, such as an employee of the organization’s network. External intruders are people outside the organization, whether external or internal. Each intruder can be involved in numerous illegal activities, working alone, as a group, or with agencies. These entities are described in detail below.

#### 4.4.1. Solo Entities

Solo entities are those that work alone with minimal safety. They are usually experts in their domains by employing a single piece of code as equipment, such as viruses, worms, and sniffers to misuse frameworks. They usually gain access to the organization’s framework through hardware damage and web loopholes. Their targets are usually little, and attacks are slightly less critical. Moreover, large and complex systems that may contain flaws are more vulnerable to attacks. Monetary institutions are also more exposed to attacks as they exchange sensitive information [72].

#### 4.4.2. Organized Groups

Solo entities are those who work alone with minimal security. They are usually experts in their field, using a single piece of code as equipment, such as viruses, worms, and sniffers to abuse frameworks. They generally gain access to the organization’s framework through hardware damage and web loopholes. Their targets are usually small, and attacks are somewhat less critical. In addition, large and complex systems that may contain flaws are more vulnerable to attack. Monetary institutions are also more exposed to attacks because they exchange sensitive information [72].

#### 4.4.3. Intelligence Agencies

Intelligent agencies from other countries are involved in this type of attack. These agencies constantly seek to test the military architecture of other nations, including contemporary monitoring and covert political and military activities. To do so, they require many resources, from software to hardware, research, development, personnel, and finances. Because they have all these resources at their disposal, some agencies now pose a serious threat to economic and military espionage. Such organizations pose the greatest threat to networks and must be closely monitored to protect the nation’s important assets [73].

### 4.5. RQ5: Network Security Attack in IoT and WSN Layers

Threats become more attractive and dangerous as technology increases. Although new security mechanisms are being developed, intruders can easily find other ways to attack systems. Table 8 explains the network security attacks in the IoT and WSN domains. The attacks are categorized according to the open system interconnection (OSI) layered point of view [74].

#### 4.5.1. Denial of Service Attacks (DOS)

A Denial-of-service (DoS) attack is a malicious attack in which attackers make the victim’s system unresponsive and difficult to reach for the legitimate user by making many requests to the expected URL than the server can handle [75,76]. DoS attacks typically occur when authenticated clients have not been granted access to the information or service [77]. A distributed denial of service (DDoS) attack is a type of DoS attack that uses multiple users or infected systems to attack a victim’s system or to attack a website so that it becomes unresponsive. This attack also prevents the website from functioning properly and disrupts regular traffic. In WSN, a DDoS attack changes the routing protocol information DSR, resulting in a huge amount of unauthorized traffic and making the network/website unavailable. On the other hand, a low-rate denial-of-service (LDoS) attack is another type of DoS that penetrates the WSN’s routing protocol, thus compromising the security and trust mechanisms. An LDoS attack is difficult to detect due to its non-stationary nature and low signal strength52. In this attack, illegitimate traffic affects the operational capability of a network. It causes severe outages and monetary losses.

#### 4.5.2. Replay Attacks

When an attacker replays a flood of messages between the sender and the receiver, updating the line table (DT) to steal information, it is called a replay attack [33]. It is a network layer attack in which a third party intercepts the data during transmission. The attacker retransmits this data by either modifying or delaying it, spoofing the sender’s IP address to the attacker’s IP address, and impersonating the legitimate sender.

#### 4.5.3. Trojan Worms, Viruses, and Malware

An attacker can use malicious software to manipulate data, steal information, or even launch a denial-of-service attack on a device. A worm, such as a Trojan horse, can infect one’s computer when one downloads a file or receives an update. The worm then multiplies and attacks other machines on the network. Unlike a virus, many Trojan horses usually reside on one’s own computer. A virus can infect the host’s file when sent via email and then spread to other users [78]. Malware is malicious content that can interfere with a computer’s operation and slow its performance. When data from IoT devices is compromised, malware can infest the cloud or data centers. These attacks breach the primary security mechanisms of any OS/server, such as a firewall and window defender.

#### 4.5.4. Black Hole Attacks

This is a network layer attack known as a packet drop attack. In this attack, a node sends an RREQ packet to all its neighbors in the network, and the router is supposed to forward the packet instead of discarding it. The nature of this attack is similar to a DDoS attack. Attackers are used to attack routers by sending many false requests to prevent legitimate routers from forwarding packets. This is also called a first-come, first-served attack because the attacker can also use a malicious router or reprogram it to block packets instead of sending correct information [78]. These attacks reduce the average throughput. When combined with a sinkhole attack, this attack affects performance and stops all traffic around the black hole. When combined with the sinkhole attack, this attack severely degrades traffic and modifies or discards content during transmission.

#### 4.5.5. Sink Hole Attacks

This is a network layer attack where attackers attract all network traffic from nearby nodes to a compromised node and appear as attractive and trusted nodes. This attack is also used to initiate other attacks such as spoofing attacks, DoS, and modification of routing information in WSN [34]. When combined with selective routing and worm attacks, sinkhole attacks become even more dangerous. A sinkhole attack is initiated in two ways, either by hacking a node within the network or by a malicious node impersonating itself as the shortest path to the base station [35,36]. A sinkhole attack impacts the routing configuration/protocols of the forwarding node. Due to this behavior, it is considered as an error or malicious node by the neighboring nodes, which affects the network performance. This leads to mis-routing and incorrect displays of the routing protocol.

#### 4.5.6. Wormhole Attacks

A wormhole attack is a network layer attack in which an invader attacks the WSN through two or more compromised nodes. The invaders forward the data from one malicious node to another node at the end of the network through the tunnel. The wormhole appears to other nodes as a fictitious neighbor. Wormhole nodes usually transmit data directly from one node to the destination without including other nodes in the path. Due to this nature, other nodes in WSN easily trust those closest to other nodes, which causes many routing problems. Moreover, they can build better communication channels for long-range communication [37]. Wormhole attacks affect the performance of many network services, such as time synchronization, localization, and data fusion.

#### 4.5.7. Selective Forwarding (Gray Hole)

A selective forwarding attack (SFA) is a special type of black hole attack in which the compromised node drops some selective packets instead of all packets. Invaders drop packets containing critical information, such as military information, without noticing them or allowing others that may contain non-critical information to pass. This can lead to worse effects and a decrease in network efficiency [34,35,36,37]. Selective forwarding attacks impact network performance and consume limited energy resources.

### 4.6. RQ6: Issues in WSN and IoT Frameworks

In this section, we mention various WSN and IoT frameworks that highlight the importance of WSN and IoT in different aspects of life. Although many advances have been made in IoT, there are still many problems, as shown in Figure 9, that need to be reduced and solved efficiently to avoid any damage [79,80].

#### 4.6.1. Security

Security is essential for any organization to protect its environment, systems, devices and applications from outside attacks. Data and communication technologies are increasing every day. Therefore, data and information security are necessary tasks [81]. In addition to data, its transmission over the network should also be protected. Although technology has evolved and security mechanisms have improved, attackers have still found many ways to breach the security level [38,39]. With the increasing number of IoT devices, new security issues have emerged. For real-time applications, the most important thing is to keep the WSN secure. The network and its associated router or hub should enforce an access control mechanism to prevent unauthorized users. Each node connected to another node is security relevant, whether it is a restricted device or a smart device. Acceptance, confirmation, categorization, trust, and information security are the most important security requirements to be considered in IoT networks and WSN. It is challenging to provide security measures for flexible detection devices. Therefore, protecting information from dictatorial forces or illegal access is called security [82,83].

#### 4.6.2. Data Confidentiality and Privacy

Data confidentiality is a significant issue in IoT and network security. In IoT frameworks, the client gains access to the information and system management in an unintended environment due to issues such as the use of sensor nodes. Attackers can physically capture them and extract data using an energy analysis attack [84]. Refurbished devices made from these captured devices can launch new attacks and violate security. Therefore, the IoT device should verify whether or not the user or device has been granted permission to access the system. The practice of controlling access to data by granting or denying permission based on a set of laws. Many devices/clients must be authenticated by management to access the system. Data confidentiality and access are the main issues in the Internet of Things (IoT). Researchers are trying to figure out how to handle the personalities of customers, items/articles, and devices in a secure manner. Due to the ubiquitous nature of IoT and WSN systems, privacy and confidentiality are major concerns in IoT devices and frameworks. Some issues, such as sniffing and spoofing, unauthorized access, data changing, forging, and unapproved alteration of IoT and WSN nodes, pose significant uncertainties in IoT. An attacker can use various IoT devices and applications to capture sensitive and personal data that is visible to outsiders.

#### 4.6.3. Data Acquisition and Transmission

The primary goal of IoT is to collect data and transmit it to where it is needed in a network. Sensors are the devices used to collect data from the environment to transmit it to the base station. After the raw data is collected, it is sent to the Sink Hub for processing. Data collection and transmission are other problems in IoT and WSN because data is exposed and modified during transmission. Data acquisition is an energy-consuming process, so extra care must be taken during gathering and transmission. Intruders can steal the data during transmission if it is not encrypted or transmitted over a secure channel. The intruder can take over a node and reprogram it with a malicious code, damaging the entire network. Therefore, security is required for this process. Sometimes intruders attack the databases of organizations to violate the confidentiality of the data [85]. Also, the intruder may destroy the node or collect important or unusual information that could be used against the system. For this reason, researchers present many security mechanisms. They protect end-to-end communication links using one-time-pad (OTP) encryption method and also identify the vulnerabilities in the DBMS application using SEPTIC method.

#### 4.6.4. Resource Limitations

If necessary resources in WSN and IoT are abandoned or not handled efficiently, it may affect the performance of the network. The network consists of many nodes and sensors that require energy to operate well [86]. Various MAC layer protocols have been developed to reduce the energy consumption of sensors or nodes. These energy-efficient algorithms work primarily by regulating the synchronization of network traffic over time and the time period during which a node becomes active in a network [87,88]. In contrast, the communication medium is another basic requirement, since nodes rely on the Internet for data transmission. There is a constant need for energy, otherwise the network will fail. The nodes have limited resources because battery capacity, correspondence capacity, and computing power are low. Again, security is the main problem, because the security measurement expenses require more resources to maintain the speed of the network, which is not affordable. As a result of low regulated security, attacks can subvert software execution and protocols used in the network [89,90].

#### 4.6.5. Quality of Service

Quality of Service (QoS) manages networks and resources to strengthen IoT connectivity. QoS manages delay, jitter, reliability, and bandwidth by classifying network traffic. It plays an important role in optimizing systems. Quality of Service means that energy efficiency, reliability, bit error rate, and latency should be good enough to capture data over a network. Therefore, it is classified in two ways: program-specific and network-specific. The QoS perspective of the network refers to the effective management of network resources and transmission performance, while the perspective of the program refers to mobility, time synchronization, and sampling parameters. Similarly, many algorithms have been developed to distribute heavy traffic evenly, and the energy consumption load in a network uses a cluster-head approach to achieve high performance and efficiency [91].

#### 4.6.6. Tampering

Sensors can be placed either indoors or outdoors. Indoor sensors can be easily managed and protected, while outdoor sensors are more vulnerable to attackers due to remote locations with poor security, harsh climates, etc 78. The probability that these sensors will be physically attacked is much higher; therefore, physical protection cannot be guaranteed. A DoS attack manipulates the network by breaking the connection or changing the current network. The attacker can also replace the original node with a fake or malicious node, causing a severe attack on the network [92]. In a Sybil attack, a malicious node penetrates each cluster head of the network and affects the operation of the routing protocol. Compromised nodes can be used to launch new attacks without exposing themselves [41,93]. These nodes are difficult to detect and isolate, allowing an attacker to alter data or transmit malware throughout the network that causes significant damage [94]. Constant monitoring of the network is necessary to ensure that WSN nodes cannot be tampered with and that network performance remains stable [95].

#### 4.6.7. Authorization and Authentication

Nodes are the building blocks of the Internet of Things that must be defined in the network. Transmission between devices and access to the entire network span a wide range in IoT and WSN. IoT devices perform role-based access control, and their devices are allowed to do only what is required [96,97]. Devices and their data must be protected from physical and logical attacks on the network. Attacks on the DNS cache could affect the overall performance of the network. Authentication is the process by which each node on the network can access data based on a fixed connection to a server or cloud-based server. If the authentication process is not administered properly, it will lead to security issues and questions. An attacker can easily access the network and make it fail temporarily by making too many wrong attempts.

Authentication is complicated due to the massive proliferation of wireless media and the nature of sensor networks. Authentication is usually done using the credentials of a legitimate user [98,99]. However, this technique is not secure enough. Therefore, passwords should be changed regularly and computers should not be left unattended to make this technique robust. Both the sender and the recipient should perform authentication to verify the origin of the communication [100,101].

### 4.7. RQ7: Limitations of the Literature Review

In this section, Table 9 explains the proposed solutions of the work conducted by various authors and the contributions with the limitations of their work are also described. The goal was to find research gaps in this area to help other researchers. The research gaps will allow researchers to develop solutions and new methods that could help fill the missing piece.

## 5. Challenges and Open Issues

Intelligent systems can address various problems faced by industry, but there have been some challenges in integrating IoT and WSN into Industry 4.0. Technological improvements in IoT and WSN have increased concerns about security and data management [96]. As more and more data is generated, it is difficult for factories and industries to manage it properly. Artificial intelligence algorithms have been implemented to manage Big Data and make systems and devices act more intelligently. The algorithms are used to process the data in different time periods. For education, the data must be shared in a central repository, while enterprises are mainly reluctant to share their private data due to poor and insufficient organizational support for data in Industry 4.0. There are also safety management issues in Industry 4.0 [116].

**Big data**: The emergence of various technologies and the explosion of their use have led to the outstanding development of Big Data technology and processing. Every device produces a huge amount of data. Due to the growing amount of big data, the improvements of Big Data packages encounter limitations and demand situations that need to be “overcome” in order to manage the amount of data used efficiently.

**Adapting to 6G**: 6G is another challenge for wireless sensor networks and the Internet of Things. Processing power is a major challenge in developing low-power and low-cost 6G devices. In addition, 6G brings privacy and security challenges for WSN and IoT.

**Updates**:system components could not be upgraded due to interoperability between protocols, systems, and their components. Therefore, systems are more vulnerable to attack if any part of a single system from a network is infected in intelligent factories.

**Environment**: security is also a critical challenge in WSN [97,100,106]. WSN nodes are not secure when deployed in a prone environment due to the wireless transmission of data. An attacker can access them from anywhere in the world and manipulate them easily. Internet attacks can also affect the vulnerability of sensor nodes.

**Supply chain management systems**: IoT devices are spreading erery day, posing new challenges to the integrity and scalability of supply chain management systems [117,118]. Simultaneously connecting IoT devices to the cloud or the Internet requires a lot of access control, fault tolerance, data management, privacy, and security.

**Limited resources**: are other challenges in WSN domain that affect the energy of sensor nodes used in the network. Sensor nodes usually change their mode from sleep to active and vice versa. Therefore, sleep mode is considered as "outside the network" while active mode brings some other issues such as energy consumption [119]. Due to the high energy consumption, they also became dead. Sensor nodes usually have limited power, processing, and memory. In addition, sensor mobility is another problem that hinders the integration of mobile sensor nodes with the Internet.

## 6. Future Directions

Industry 4.0 leads to the merging of people and technology to complement human activities with intelligent machines. Industry 4.0 will lead to customized human fashion that will minimize the oversupply and unavailability of supplies or items. Human-machine interaction will increase productivity and customer satisfaction with customized products.

The next version of Industry 4.0 is Industry 5.0, which is expected to be more user-friendly and better integrate technology with society and the environment. It depends mainly on robots. Robots are already the backbone of manufacturing, and Industry 4.0 technologies [120,121] provide flexibility in manufacturing. Industry 5.0 combines human creativity and craftsmanship with the speed, productivity (e.g., CPS) [122], and consistency of robots. In this version, robots can be programmed to work alongside humans.

**Soft computing**: can be used to reduce the dimensions of these large dimensional data sets [123]. Good features are essential to make efficient decisions. This is the reason why soft computing techniques are used to obtain useful features.

**Explainable artificial intelligence (XAI)**: can be used for interpretability of the decision made by the classification model. Classification models make decisions in a black box where the user does not know how the decision is made. XAI converts this black box into a white box and interprets the decision made by a model. XAI increases user confidence to take further action [124].

**Federated learning (FL):** is an optimal choice for privacy preservation. FL works by training global and local models on the edge device. The model on the edge device does not share the data with the global, thus keeping the data private at each edge device. Only parameters are shared globally to retrain the global model and optimize the inference results [125,126].

**Secure devices**: sensor nodes are designed to consume less energy and become active when they are needed or an event occurs [59]. Further improvements are also needed to prevent attacks from the Internet. While the IoT has no limitations in terms of processing or energy. Due to the tremendous proliferation of IoT devices, this paradigm is now being shifted from the IoT to the Internet of Everything.

**Sustainability**: IoT systems are now moving toward the idea of self-organization, and systems are becoming capable of responding in an automated and adaptive manner and dealing with changes and uncertainties in the environment [118].

**Education 5.0**: in this digital era, education must also change from traditional to integrating hardware and software with co-bots to develop new skills and a smart society. Educational institutes are now using pedagogical tools to provide a better experience. Even though IoT-based education is still not widespread, there is still room for further improvement, such as sensor node coverage and efficiency, wireless data transmission of data [127], battery life, and high-cost nodes.

**General directions:** there are many challenges. Future directions may address heterogeneous interoperability of systems, self-organization protocols, routing schemes for managing IoT networks, data management [79], cross-platform optimization, and the development of network security algorithms to secure wireless transmission from data manipulation, stealing, or hacking activities. On the hardware side, researchers are developing energy-efficient sensor nodes [91,115], with net-zero power to reduce maximum power consumption.

## 7. Conclusions

In this digital and modern era, technology is evolving every day. Due to the massive proliferation of technology, IoT and WSN play an important role in Industry 4.0 to develop smart applications, design networked data centers, and build autonomous smart industries. Data networks have been created and improved with the help of new and smart devices. In this systematic literature review, WSN and IoT network threats were analyzed and a descriptive comparative study was conducted. These networks are the main attack surfaces for attackers to draw meaningful patterns from system and user data. Wireless sensor networks (WSN) and the Internet of Things (IoT) have rapidly (and widely) evolved to meet the increasing demand for conventional application scenarios, such as plant automation and remote process control systems. These smart devices are also being used to improve the efficiency of existing networks and create new opportunities for automating and securing industrial processes. In this article, we explore seven research questions: (i) What are the contributions of WSN in IR4. 0? (ii) What are the contributions of IoT in IR 4.0? (iii) What are the types of WSN coverage areas for IR 4.0? (iv) What are the main types of network intrusions in WSN and IoT systems? (v) What are the main network security attacks in WSN and IoT? (vi) What are the major issues in IoT and WSN systems? (vii) What are the limitations and research gaps in the current work? The main purpose of the fourth industrial revolution with WSN and IoT explicitly shows that the evolutionary transition needs to be intensified and extended to include emerging research areas and intimidating technological challenges. This article covers all elements of WSN, from the design phase to the security requirements, from the implementation phase to the classification of the network, and from the difficulties and challenges of WSN. Future studies will address the problems in the coverage regions of wireless sensor networks and provide effective solutions to the existing problems and challenges in this area. The use and application of WSN and IoT in Industry 4.0 involves the processing of extracted data and the efficient and secure transmission of this data to a remote location.

## Figures and Tables

**Figure 1 sensors-22-02087-f001:**
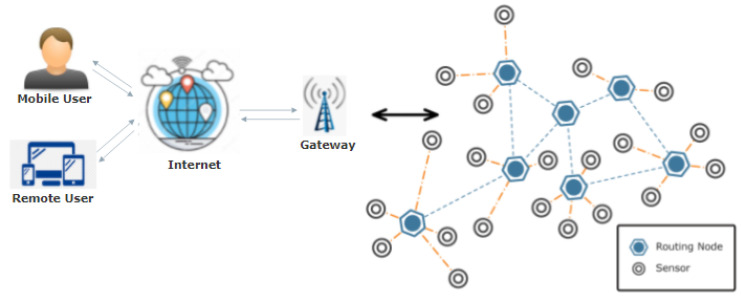
Architecture of wireless sensor network (WSN).

**Figure 2 sensors-22-02087-f002:**
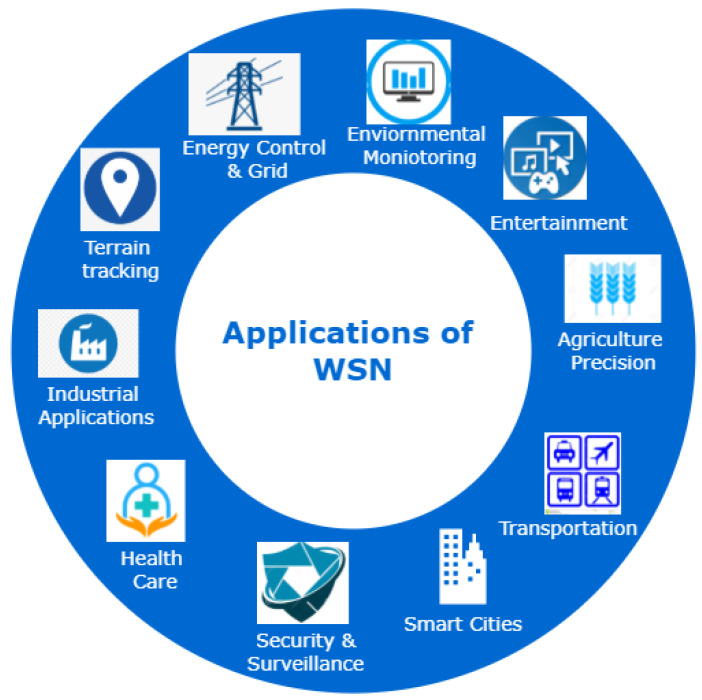
Application of wireless sensor network (WSN).

**Figure 3 sensors-22-02087-f003:**
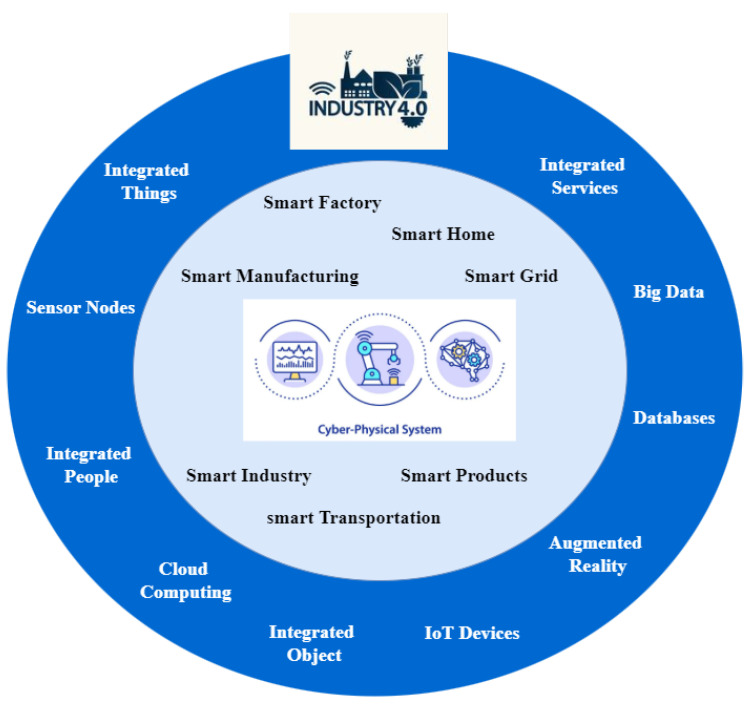
CPS system–integration of IoT, wireless devices, and people in Industry 4.0.

**Figure 4 sensors-22-02087-f004:**

Planning process of systematic literature review (SLR).

**Figure 5 sensors-22-02087-f005:**
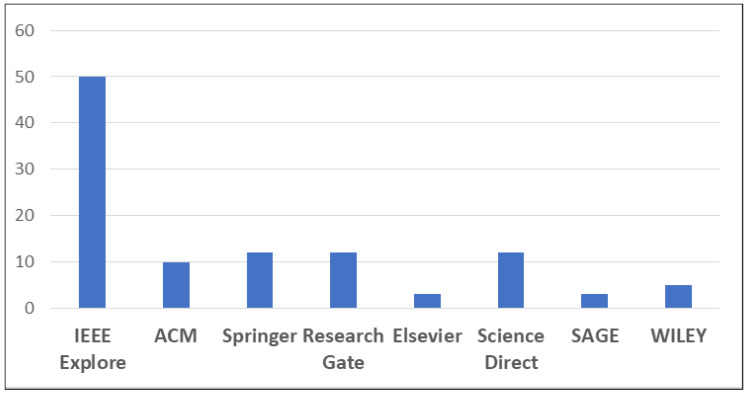
Repositories versus number of studies Used.

**Figure 6 sensors-22-02087-f006:**
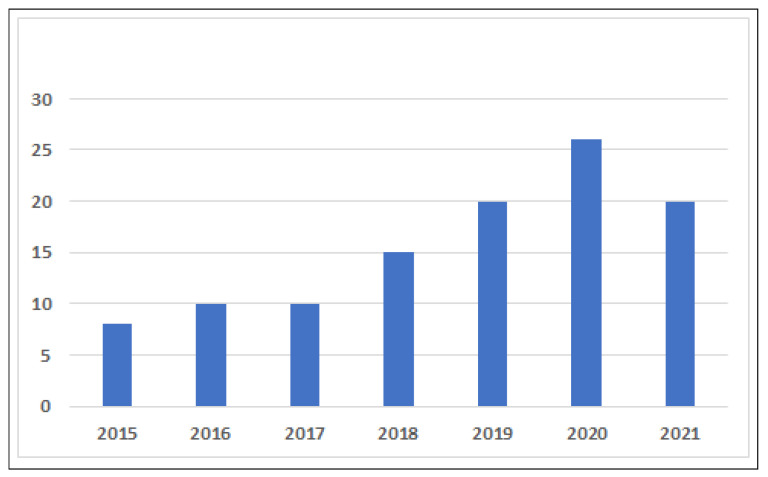
Year-wise distribution of articles.

**Figure 7 sensors-22-02087-f007:**
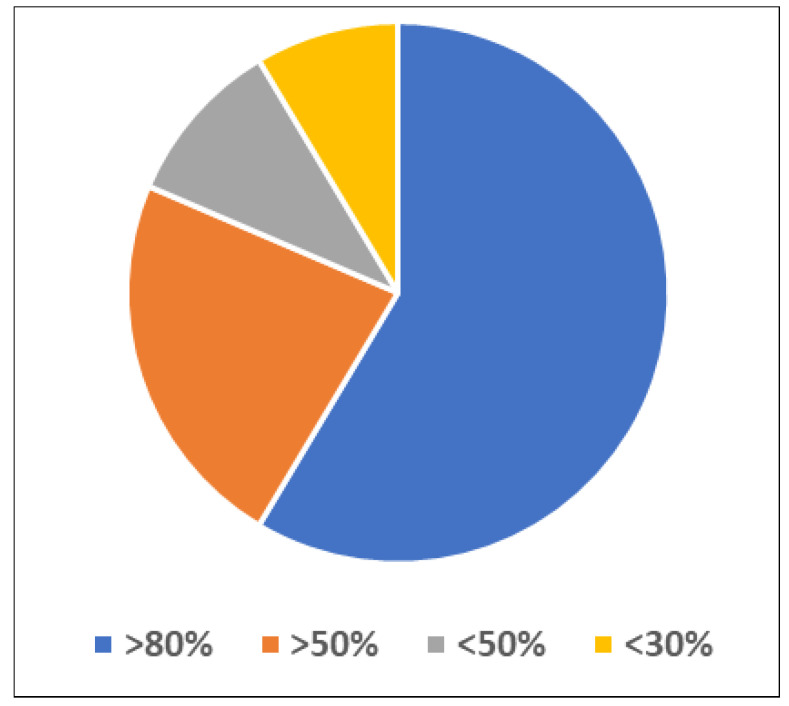
Graded percentage of selected studies.

**Figure 8 sensors-22-02087-f008:**
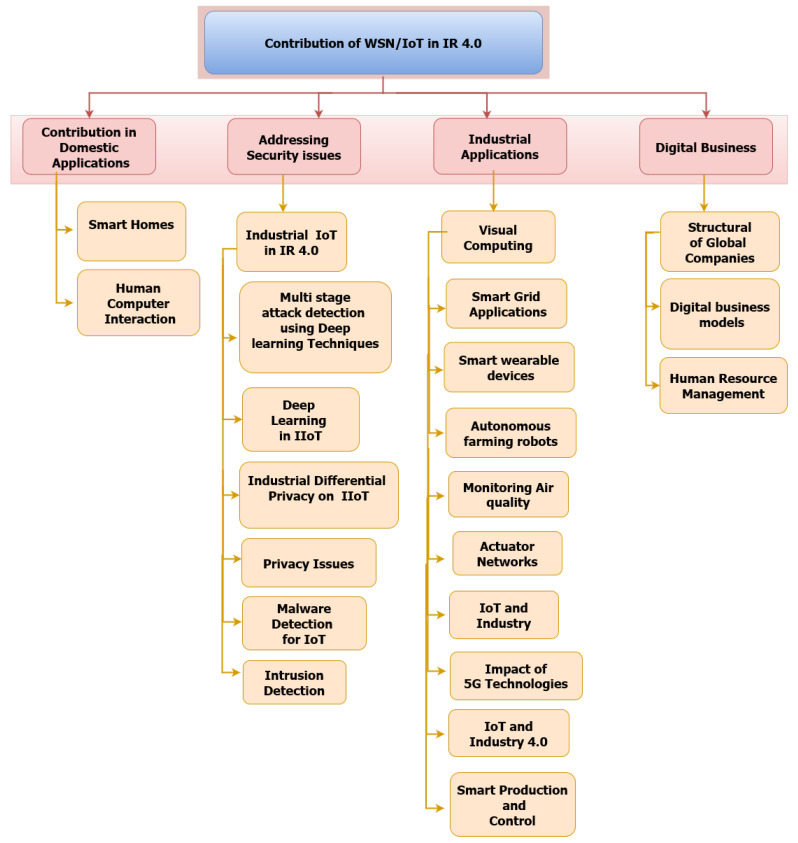
Taxonomy of existing studies.

**Figure 9 sensors-22-02087-f009:**
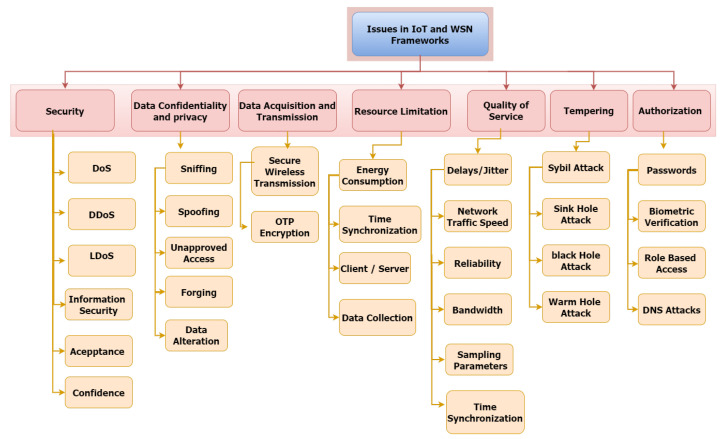
Issues in WSN and IoT Framework.

**Table 1 sensors-22-02087-t001:** List of Abbreviations.

Abbreviation	Description
5G	fifth generation
6G	sixth generation
AR	augmented reality
CPS	cyber physical system
DDoS	distributed denial of service
DNS	domain name system
DoS	denial of service
DT	digital twin
FoI	field of interest
GUI	graphical user interface
ID 4.0	Industry 4.0
IIoT	industrial internet of things
IoT	internet of things
IR 4.0	industry revolution 4.0
IWSN	industrial wireless sensor network
IWSAN	industrial wireless sensor and actuator network
PREQ	request packet
QoS	quality of service
RFID	radio frequency identification
SEPTIC	self protecting databases from attacks
SG	smart grid
SIRP	self-optimized smart routing protocol
SLR	systematic literature review
UASN	underwater acoustic sensor networks
WSN	wireless sensor network
WSAN	wireless sensor area network

**Table 2 sensors-22-02087-t002:** Comparative analysis of the existing review papers. Key: deployment category—DC, literature review—LR, security overview—SO, bibliometric literature review—BLR, systematic literature review—SLR.

Reference	Year	Review Type	DC	Application Types	IoT and WSN Architecture Used	Challenges and Issues	R. Q.
[1]	2015	LR	Industry	802.11 (WiFi) technology in smart cities	WSN	×	×
[4]	2020	LR	Science	IoT sensing applications discussed using sensing technology	WSN using RFID	Energy harvesting, communication interference, fault tolerance, higher capacities to handling data processing, cost feasibility.	×
[5]	2016	SO	Industry	IoT application in smart grids	IoT	Challenges discussed along with solutions to cope with	×
[6]	2017	SO	Industry	Deployment techniques discussed using sensor network	WSN	Communication cost, coverage time, accuracy, etc.	×
[10]	2016	LR	WSN applications in urban areas	Urban areas	WSN	Problems and solution of each WSN application	×
[17]	2014	SO	Industry	Network security protocols discussed in industrial applications	WSN	Challenges of stack protocol and their solutions	×
[19]	2020	SLR	Smart factories	Scope and conceptualization of IoT in Industry 4.0	IoT	×	✔
[27]	2020	LR	Smart IoT devices	Detailed survey on security threat models applicable for IoT and WSN. They also discussed communication attacks and taxonomy of IoT and WSN	Both	✔	×
[28]	2019	LR	–	Discussed technical and social perspective of IoT for future technology enhancement	IoT	✔	×
[29]	2017	SLR	Smart cities	Applications, security, and taxonomy in IoT	IoT	×	×
[30]	2019	LR	Industrial	Applications and usage of actuators and sensor networks using MAC protocol.	IWSN	Security challenges on different layers of the stack, also discussed their solutions	×
[31]	2016	LR	–	Technologies, innovations, and applications of IoT discussed.	IoT	✔	×
[32]	2014	LR	Industrial	Coverage areas of WSN are discussed	WSN	Challenges they face were: Node type, depth type, communication range, etc.	×
[33]	2016	SUR	Industrial	Applications of intrusion detection system in IoT	IoT	×	×
[34]	2015	SUR	Industrial	Only explore and analyze existing solution to detect sinkhole attack	WSN	×	×
[35]	2020	LR	–	×	Both	Discuss attacks in IoT ans WSN with their solutions, advantages, and limitation	×
[36]	2021	SLR	Smart mobiles	Routing attacks and security measures in mobile network are discussed	WSN	×	×
[37]	2021	LR	Industrial	Detection of wormhole in both domains	Both	×	✔
[38]	2017	SUR	IoT systems	Software board and chips, crypto algorithms, security of IoT systems, and network protocols are discussed	IoT	–	×
[39]	2015	SUR	–	Existing security approaches of IoT system are described	IoT	✔	×
[40]	2016	SUR	–	Deployment models for sensor network to achieve coverage, their classification and working was discussed	WSN	×	×
[41]	2018	BLR	Smart factory & Industry	Discuss 12 approaches of Industry 4.0. in business and account management fields	IoT	×	×
This Paper	2021	SLR	Smart industry and Factory	Applications and contribution of both IoT and WSN are discussed in detail	IoT and WSN (both)	Key challenges and open issues of both IoT and WSN in Industry 4.0 are discussed	✔

**Table 3 sensors-22-02087-t003:** Research questions.

	Research Questions (RQ)
RQ1	What are the contributions of WSN in IR 4.0?
RQ2	What are the contributions of IoT in IR 4.0?
RQ3	What are the types of WSN coverage areas for IR 4.0?
RQ4	What are the major types of network intruders in WSN and IoT systems?
RQ5	What are the prominent network security attacks in WSN and IoT?
RQ6	What are the major issues in IoT and WSN frameworks?
RQ7	What are the limitations and research gaps in the existing work?

**Table 4 sensors-22-02087-t004:** Search strings.

Sr. No.	Groups	Group Search Query
1	Group 1	Application of WSN in IR 4.0.
2	Group 2	Implementation of IoT infrastructure in IR 4.0.
3	Group 3	Industrial Revolution 4.0. for smart manufacturing
4	Group 4	Security attacks, issues, and challenges of IoT and WSN in IR 4.0.
5	Group 5	Role of WSN and IoT systems in IR 4.0.

**Table 5 sensors-22-02087-t005:** Inclusion and exclusion criteria.

Inclusion Criteria	
1	Include only those papers written in the English language.
2	Include papers that were published in 2014–2021.
3	Include papers that reflected enough knowledge about the search strings and search objectives.
4	Include papers whose titles, keywords, abstracts, and conclusions provided enough information related to WSN, IoT, and IR 4.0.
5	Include papers whose content focused on WSN, IoT, and IR 4.0 content and provided in depth insights.
Exclusion Criteria	
1	Exclude papers written in a language other than the English language.
2	Exclude gray papers.
3	Exclude papers that were not published within 2014–2021.
4	Exclude research papers containing less than three pages.
5	Exclude papers that failed to meet the inclusion criteria.

**Table 6 sensors-22-02087-t006:** Selection criteria versus response graded.

Criteria	Selection Criteria	Graded Response
C1	Is the aim of research and context clearly defined?	1, 0.5, 0 (yes, nominally, no)
C2	Is the context of research well addressed?	1, 0.5, 0 (yes, nominally, no)
C3	Are the findings clearly stated?	1, 0.5, 0 (yes, nominally, no)
C4	Based on the findings, how valuable is the research?	>80% = 1, <20% = 0, in between = 0.5

**Table 7 sensors-22-02087-t007:** Selected studies used for SLR analysis.

Sr. No.	Title of Research	Authors	Year
1.	Cyber-Physical Systems Security: Analysis, Challenges, and Solutions	Y. Ashibani and Q. H. Mahmoud	2017
2.	A Review of IoT sensing applications and challenges using RFID and wireless sensor networks	H. Landaluce, L. Arjona, A. Perallos, F. Falcone, I. Angulo, and F. Muralter	2020
3.	Enhancement of relay nodes communication approach in WSN-IoT for underground coal mine	R. Sharma and S. Prakash	2020
4.	Applications of wireless sensor networks for urban areas: A survey	B. Rashid and M. H. Rehmani	2016
5.	An empirical study of application layer protocols for IoT	U. Tandale, B. Momin, and D. P. Seetharam	2017
6.	Digital twin technologies and smart cities.	M. Farsi, A. Daneshkhah, A. Hosseinian-Far, and H. Jahankhani	2020
7.	Internet of things (IoT) embedded future supply chains for Industry 4.0: An assessment from ERP-based fashion apparel and footwear industry	M. A. A. Majeed and T. D. Rupasinghe	2017
8.	Towards Industry 4.0 utilizing data-mining techniques: a case study on quality improvement	H. Oliff and Y. Liu	2017
9.	An industrial perspective on wireless sensor networks-a survey of requirements, protocols, and challenges	A. A. Kumar S., K. Ovsthus, and L. M. Kristensen.	2014
10.	The smart factory as a key construct of Industry 4.0: A systematic literature review	P. Osterrieder, L. Budde, and T. Friedli	2020
11.	Social expectations and market changes in the context of developing the Industry 4.0 concept	S. Saniuk, S. Grabowska, and B. Gajdzik	2020
12.	Key IoT Statistics	B. Jovanović	2021
13.	30 Internet of Things – IoT stats from reputable sources in 2021	A. Multiple	2021
14.	Wide-area and short-range IoT devices	S. O’Dea	2021
15.	The Future of Industrial Communication: Automation Networks in the Era of the Internet of Things and Industry 4.0	M. Wollschlaeger and T. Sauter and J. Jasperneite	2017
16.	Internet of things (IoT): a technological analysis and survey on vision, concepts, challenges, innovation directions, technologies, and applications	G. Misra, V. Kumar, A. Agarwal, and K. Agarwal	2016
17.	EDHRP: Energy-efficient event-driven hybrid routing protocol for densely deployed wireless sensor networks	Faheem M, Abbas MZ, Tuna G, Gungor VC.	2015
18.	A survey on deployment techniques, localization algorithms, and research challenges for underwater acoustic sensor networks.	Tuna G, Gungor VC	2017
19.	Lrp: Link quality-aware queue-based spectral clustering routing protocol for underwater acoustic sensor networks	Faheem M, Tuna G, Gungor VC	2017
20.	Design and deployment of a smart system for data gathering in aquaculture tanks using wireless sensor networks	Parra L, Sendra S, Lloret J, Rodrigues JJ.	2017
21.	WSN-and IoT-based smart homes and their extension to intelligent buildings. Sensors	Ghayvat H, Mukhopadhyay S, Gui X, Suryadevara N	2015
22.	Conceptual model for informing user with an innovative smart wearable device in Industry 4.0	M. Periša, T. M. Kuljanić, I. Cvitić, and P. Kolarovszki	2019
23.	Evolution of wireless sensor network for air quality measurements	Arroyo, P.; Lozano, J.; Suárez, J.	2018
24.	Industrial wireless sensor and actuator networks in Industry 4.0: Exploring requirements, protocols, and challenges—A MAC survey	S. Raza, M. Faheem, and M. Genes	2019
25.	Cause the Industry 4.0 in the automated industry to new requirements on the user interface	C. Wittenberg	2015
26.	Impact of 5G technologies on Industry 4.0	G. S. Rao and R. Prasad	2018
27.	Material efficiency in manufacturing: Swedish evidence on potential, barriers, and strategies	S. Shahbazi et al.	2016
28.	Organizational change, and industry 4.0 (id4). A perspective on possible future challenges for human resources management	J. Radel	2017
29.	Organizational culture as an indication of readiness to implement Industry 4.0	Z. Nafchi and M.Mohelská	2020
30.	Smart production planning and control: concept, use-cases, and sustainability implications	O.E, Oluyisola	2020
31.	Fortune favors the prepared: How SMEs approach business model innovations in Industry 4.0.	J. M. Müller et al.	2018
32.	Visual computing as a critical enabling technology for industries 4.0 and industrial Internet	J. Posada et al.	2015
33.	Digitalization and energy consumption. Does ICT reduce energy demand	S. Lange	2020
34.	Industry 4.0: adoption challenges and benefits for SMEs	T. Masood and P. Sonntag	2020
35.	Measurement and analysis of corporate operating vitality in the age of digital business models	J. Zhu et al.	2020
36.	Cyber security and the Internet of Things: vulnerabilities, threats, intruders and attacks	M. Abomhara and G. M. Køien	2015
37.	Sharing user IoT devices in the cloud	Y. Benazzouz, C. Munilla O. Gunalp, M. Gallissot, and L. Gurgen	2014
38.	Security in Internet of things: Challenges, solutions, and future directions	S. A. Kumar, T. Vealey, and H. Srivastava	2016
39.	Survey of intrusion detection system towards an end-to-end secure internet of things	A. A. Gendreau, M. Moorman	2016
40.	Recent advances and trends in predictive manufacturing systems in a big data environment	J. Lee et al.	2015
41.	A comprehensive dependability model for QOM-aware industrial WSN when performing visual area coverage in occluded scenarios	T. C. Jesus, P. Portugal, D. G. Costa, and F. Vasques	2020
42.	Security issues and challenges on wireless sensor networks	M. A. Elsadig, A. Altigani, and M. A. A. Baraka	2019
43.	Challenges of Wireless Sensor Networks and Issues associated with Time Synchronization	G. S. Karthik and A. A. Kumar	2015
44.	Design and analysis of intrusion detection protocols for hierarchical wireless sensor networks	M. Wazid	2017
45.	Intrusion detection protocols in wireless sensor networks integrated to the Internet of Things deployment: survey and future challenges	S. Pundir, M. Wazid, D. P. Singh, A. K. Das, J. J. P. C. Rodrigues, and Y. Park	2020
46.	Robust malware detection for Internet of (battlefield) Things devices using deep Eigenspace learning [46]	Azmoodeh, A. Dehghantanha, and K.-K.-R. Choo	2019
47.	LSDAR: A lightweight structure-based data aggregation routing protocol with secure IoT integrated next-generation sensor networks.	Haseeb K, Islam N, Saba T, Rehman A, Mehmood Z.	2020
48.	SEPTIC: Detecting injection attacks and vulnerabilities inside the DBMS.	Medeiros, M. Beatriz, N. Neves, and M. Correia	2019
49.	An efficient ECC-based provably secure three-factor user authentication and key agreement protocol for wireless healthcare sensor networks. Computers and Electrical Engineering	Challa S, Das AK, Odelu V, Kumar N, Kumari S, Khan MK, et al.	2018
50.	Internet of Things: vision, applications and challenges	Rishika Mehta, Jyoti Sahnib, Kavita Khannac	2018
51.	A roadmap for security challenges in the Internet of Things	Arabia Riahi Sfar, Enrico Natalizio, Yacine Challal, Zied Chtourou	2018
52.	A novel low-rate denial of service attack detection approach in ZigBee wireless sensor network by combining Hilbert-Huang transformation and trust evaluation	H. Chen, C. Meng, Z. Shan, Z. Fu, and B. K. Bhargava	2019
53.	Analysis of quantities influencing the performance of time synchronization based on linear regression in low-cost WSN	D. Capriglione, D. Casinelli, and L. Ferrigno	2016
54.	C–Sync: Counter-based synchronization for duty-cycled wireless sensor networks	K.-P. Ng, C. Tsimenidis, and W. L. Woo	2017
55.	Time synchronization in WSN with random bounded communication delays.	Y.-P. Tian	2017
56.	A novel model of Sybil attack in cluster-based wireless sensor networks and propose a distributed algorithm to defend It	M. Jamshidi, E. Zangeneh, M. Esnaashari, A. M. Darwesh, and A. J. Meybodi	2019
57.	Challenges, threats, security issues, and new trends of underwater wireless sensor networks	G. Yang, L. Dai, and Z. Wei	2018
58.	Industry 4.0 key research topics: A bibliometric review	D. Trotta and P. Garengo	2018
59.	Privacy in the Internet of Things: threats and challenges	J. H. Ziegeldorf, O. G. Morchon, and K. Wehrl	2015
60.	On the security and privacy of the Internet of Things architectures and systems.	E. Vasilomanolakis, J. Daubert, M. Luthra, V. Gazis, A. Wiesmaier and P. Kikiras	2015
61.	Cybersecurity issues in wireless sensor networks: current challenges and solutions	D. E. Boubiche, S. Athmani, S. Boubiche, and H. Toral-Cruz	2020
62.	A security model for IoT-based systems	Z. Safdar, S. Farid, M. Pasha, and K. Safdar	2017
63.	Security issues and challenges in IoT routing over wireless communication	G. Saibabu, A. Jain, and V. K. Sharma	2020
64.	Security and privacy consideration for Internet of Things in smart home environments	Desai, Drushti, and Hardik Upadhyay	2015
65.	E.D. Security and grand privacy challenges for the Internet of Things	Fink, G.A., Zarzhitsky, D. V., Carroll, T.E., and Farquhar	2015
66.	A comprehensive approach to privacy in the cloud-based Internet of Things.	Henze, M., Hermerschmidt, L., Kerpen, D., Häußling, R., Rumpe, B., and Wehrle, K.	2016
67.	Towards an analysis of security issues, challenges, and open problems on the internet of Things.	Hossain, A. J., Fotouhi, M., and Hasan, R.	2015
68.	An End-to-end view of IoT security and privacy	Zhen Ling, Kaizheng Liu, Yiling Xu, YierJin, XinwenFu	2017
69.	Security and privacy considerations for IoT application on smart grids: Survey and research challenges	Dalipi, F.; Yayilgan, S.Y.	2016
70.	Internet of Things security: A survey	Alaba, Fadele Ayotunde, et al.	2017
71.	Security for the Internet of things: a survey of existing protocols and open research issues	J. Granjal, E. Monteiro, J. Silva	2015
72.	Security, privacy and trust in Internet of things: the road ahead	S. Sicari, A. Rizzardi, L.A. Grieco, A. Coen-Porisini	2015
73.	Access control and authentication in the Internet of Things environment	A.K. Ranjan, G. Somani	2016
74.	Toward secure and provable authentication for the Internet of Things: realizing Industry 4.0	S. Garg, K. Kaur, G. Kaddoum, and K. K. R. Choo	2020
75.	Prediction of satellite shadowing in smart cities with application to IoT	S. Hornillo-Mellado, R. Martín-Clemente, and V. Baena-Lecuyer	2020
76.	Software-defined industrial Internet of Things in the context of Industry 4.0	J. Wan et al.	2016
77.	Residual energy-based cluster-head selection in WSN for IoT application.	T. M. Behera, G. S. Mohapatra, U. C. Samal, M. G. S. Han, M. Daneshmand, and A. H. Gandomi	2019
78.	DistB-SDoIndustry: enhancing security in Industry 4.0 services based on the distributed blockchain through software-defined networking-IoT enabled architecture,	A. Rahman et al.	2020
79.	Application of IoT-aided simulation to manufacturing systems in the cyber-physical system	Y. Tan, W. Yang, K. Yoshida, and S. Takakuwa	2019
80.	Convergence of blockchain and edge computing for secure and scalable IIoT critical infrastructures in Industry 4.0 [47]	Y. Wu, H.-N. Dai, and H. Wang	2020
81.	Comparative study of IoT-based topology maintenance protocol in a wireless sensor network for structural health monitoring	M. E. Haque, M. Asikuzzaman, I. U. Khan, I. H. Ra, M. S. Hossain, and S. B. Hussain Shah	2020
82.	Toward dynamic resources management for IoT-based manufacturing	J. Wan et al.	2018
83.	SENET: A novel architecture for IoT-based body sensor networks	Z. Arabi Bulaghi, A. Habibi Zad Navin, M. Hosseinzadeh, and A. Rezaee	2020
84.	Bio-inspired routing protocol for WSN-based smart grid applications in the context of Industry 4.0	M. Faheem et al.	2019
85.	IoT and wireless sensor network-based autonomous farming robot	A. Khan, S. Aziz, M. Bashir, and M. U. Khan	2020
86.	Efficient and secure three-party mutual authentication key agreement protocol for WSN in IoT environments	C. T. Chen, C. C. Lee, and I. C. Lin	2020
87.	Wireless sensor network combined with cloud computing for air quality monitoring	P. Arroyo, J. L. Herrero, J. I. Suárez, and J. Lozano	2019
88.	Edge computing-enabled wireless sensor networks for multiple data collection tasks in Smart Agriculture	X. Li, L. Zhu, X. Chu, and H. Fu	2020
89.	Cluster centroid-based energy-efficient routing protocol for WSN-Assisted IoT	N. Prophess, R. Kumar, and J. B. Gnanadhas	2020
90.	An energy-efficient and secure IoT-based WSN framework: an application to smart agriculture	K. Haseeb, I. U. Din, A. Almogren, and N. Islam	2020
91.	Deployment schemes in a wireless sensor network to achieve blanket coverage in large-scale open area	Vikrant Sharmaa, R.B. Patelb, H.S. Bhadauriaa, D. Prasadc	2016

**Table 8 sensors-22-02087-t008:** Network Security Attacks on IoT and WSN Layers.

Sr. No.	Layer Name	Attacks
1	Physical layer	Interception, radio interference, jamming, tempering, Sybil attack.
2	Data link layer	Replay attack, Spoofing, altering routing attack, Sybil Attack, collision, traffic analysis, and monitoring, exhaustion.
3	Network layer	Black hole attack, wormhole attack, sinkhole attack, grey hole attack, selective forwarding attack, hello flood attack, misdirection attack, internet smurf attack, spoofing attack.
4	Transport layer	De-synchronization, transport layer flooding attack.
5	Application layer	Spoofing, alter routing attack, false data ejection, path-based DoS.

**Table 9 sensors-22-02087-t009:** Contributions and limitation of the literature.

Reference	Title of Article	Proposed Solution	Limitations and Future Work
Sharma et al. [9]	Enhancement of relay nodes communication approach in WSN-IoT for underground coal mine	They designed relay node structures for a wireless sensor network and load balancing to improve network lifetime parameters. They designed an IoT-based WSN to provide advance warning of any natural disaster in coal mines.	There were several analysis parameters to analyze the networks, such as network lifetime, communication and transmission cost, energy consumption, and coverage of the whole area.
Faheem et al. [49]	Bio-inspired routing protocol for WSN-based smart grid applications in the context of Industry 4.0	They designed a comprehensive, optimized, and QoS monitoring multi-hop network system for real-time data transmission in Industry 4.0. This self-optimized smart routing protocol (SIRP) was efficiently used for WSN-based SG applications.	In the future, they will attempt to enhance their developed SIRP routing scheme and communications architecture to collect QoS-aware data for different WSN-based smart grid applications with little data redundancy.
Arslan et al. [52]	IoT and wireless sensor network-based autonomous farming robot	They developed a computer vision-based algorithm used for the classification of weed and a non-image. Wireless sensor nodes detect weed images through image processing methods and gather light, temperature, humidity, and moisture data.	The limitation of this work is that they did not provide any GUI or mobile application control to work robot autonomously.
Chen et al. [53]	Efficient and secure three-party mutual authentication key agreement protocol for WSN in IoT environments	They proposed a practical and secure approach to merge IoT and WSN. Their scheme had high performance, low communication, and computational costs, low energy consumption, and provided effective authentication of the user in IoT.	The limitation of this study is that they did not provide a solution to the security threats in a heterogeneous IoT environment. In the future, they will evaluate the reliability and scalability of their systems of heterogeneous environments.
Rathee et al. [102]	A secure IoT sensors communication in Industry 4.0 using blockchain technology	Wireless sensor network security improved using blockchain and compared security metrics. &It ensured confidentiality and responsibility and tracked each sensor’s operation. The blockchain was used to store IoT artifacts and sensors.	The developed IoT sensor takes time to test a single block before it is put to the blockchain.
Mellado et al. [103]	Prediction of satellite shadowing in smart cities with application to IoT	The technology had a minimal processing load. It was highly desirable to create a coverage map that can optimize network resources in satellites.	There is a lack of evaluation of requirements for satellite-based IoT and output connectivity protocols through simulations in actual situations.
Garg et al. [101]	Towards secure and provable authentication for the internet of things: realizing Industry 4.0	The effectiveness of the developed protocol was evaluated with frequently utilized AVISPA, PUFs, and ECC encryption algorithms. A proposed technique was developed to create a durable, stable, and efficient user architecture that promotes shared authentication for IoT and server nodes and is resistant to cyber threats.	This protocol is for academic and research purposes only, and its implementation has not yet been tested in the real world.
Behera et al. [104]	Residual energy-based cluster-head selection in WSN for IoT application	The method takes into account the intended value of initial energy, residual energy, and cluster heads to choose the specific set of cluster heads in the network that adapts IoT applications to maximize flow, durability, and residual energy.	They did not review existing path selection factors in a node mobility network that altered its role constantly.
Wan et al. [105]	Software-defined industrial Internet of Things in the context of industry 4.0	They proposed a new idea of information interaction in Industry 4.0 using software-defined IIoT. They enhanced the network size using IIoT. The IIoT architecture manages physical devices and information exchange methods via a customized networking protocol.	The limitation of the study is the effective coordination between IIoT where the network is heterogeneous for transmission of information.
Tan et al. [106]	Application of IoT-aided simulation to manufacturing systems in cyber-physical systems	They discussed the construction and implementation methods of digital twin (DT). In this study also explained the issues involved in developing DT with the help of IoT manufacturing devices. DT is the simulation tool that can gather and synchronize data for the real world to a real-time environment.	The absence of experimentation and optimization in predicting future locations or results are other essential aspects of DT.
Rahman et al. [107]	DistB-SDoIndustry: enhancing security in Industry 4.0 services based on the distributed blockchain through software-defined networking-IoT enabled architecture	In this work, the authors develop a distributed blockchain-based security system integrated with the help of IoT and SDN. Blockchain is used for data security and confidentiality, while SDN-IoT incorporates sensor networks and IoT devices to improve the security services in Industry 4.0.	Limitations of this study are that the developed model SDN-IoT was still in the initial stage, so it was not able to detect different types of risks, such as service denial (DoS) and flood attack and packet filtering. The developed system had no proper GUI, so the throughput, packet arrival time, and response time were rarely challenging to analyze.
Haque et al. [108]	Comparative study of IoT-based topology maintenance protocol in a wireless sensor network for structural health monitoring	They developed a computer-based monitoring system to analyze the vibration or earthquake measurement. WSN are used to sense structural damages and identify their pinpoint location. They also proposed a topology-based maintenance system to analyze network architecture. Their system was an energy-efficient system that automatically turned off nodes where no traffic was detected.	The limitation of this study is that WSN nodes are not capable enough to provide scalability for large coverage areas.
Wan et al. [109]	Toward dynamic resources management for IoT-based manufacturing	To build a fully interactive environment and dynamic management of resources, an ontology-based technology, SDN, communication technology device to device combined with ontology modeling and multi-agency technology were used to accomplish sophisticated administration of resources. They solved load secluding problems using Jena logic reasoning and contract-net protocol-based technology in Industry 4.0.	The limitation of this work was the high time complexity of the load balancing algorithm to complete the task efficiently. It was challenging to refine the process due to the complex nature of multi-agent technology, and referencing rules were much more complex.
Bulaghi et al. [110]	SENET: a novel architecture for IoT-based body sensor networks	Multiple algorithms, such as particle swarm optimization (PSO), ant colony optimization (ACO), and genetic algorithms (GA) were used to save energy of WSN. They evaluated WSN energy consumption using optimization algorithms and calculated the total number of uncovered points, their stability, and dependability.	The design meets some disadvantages and does not work in real-time data.
Thiago et al. [111]	A comprehensive dependability model for QoM-aware Industrial WSN	When performing visual area coverage in occluded scenarios. They proposed a mathematical model named quality of monitoring parameter (QoM) to assess the dependability of WSN, their availability, and reliability considering hardware, networking, and visual coverage failures.	Their developed method was inefficient at analyzing the system’s dependability in real-time applications due to failures or repairs happening as soon.
Patricia et al. [112]	Wireless sensor network combined with cloud computing for air quality monitoring	They designed a small size, low cost, and efficient system to monitor the air quality using wireless sensor nodes. They performed multiple algorithms such as multi-layer perceptron, SVM, and PCA to discriminate and quantify the volatile organic compounds.	The limitation of this study is that sensor nodes are less efficient at covering a large area to monitor and cannot do real-time testing and the field measurements of sensors.
Li et al. [113]	Edge computing-enabled wireless sensor networks for multiple data collection tasks in smart agriculture	They designed a data collection algorithm considering data quality factors in smart agriculture. Then modeled the data collection process by merging WSN and IoT.	The developed edge computing driven framework [47] and data collection algorithm were not capable of collecting data in a real agriculture environment.
Kumar et al. [114]	Cluster centroid-based energy-efficient routing protocol for WSN-assisted IoT	They developed a system that was capable of self-organization of local nodes to save energy. Their system adopted new algorithms to rotate head clusters based on centroid locations in IoT using WSN. The technique exceeds conventional protocols for efficiency criteria, such as the consumption of energy by the network, intermediate sensor node, packet distribution ratio, packet failure percentage, and network output. Their work was best for the base station located in the network.	The routing protocol was not optimal, routing strategies were lacking, and packet loss was caused if the base stations were even in the network. In the future, they will enhance this work by using a multi-hop path strategy to the base station. In this technique, the cluster head will transmit data to the base station, even outside the network.
Haseeb et al. [115]	An energy-efficient and secure IoT-based WSN framework: an application to smart agriculture	They proposed an IoT-based WSN framework that collected data from agriculture and transmitted it to the nearest base station. They enhanced network throughput, low latency rate, energy consumption, and packet drop ratio. They also provided security to the data transmission channel using the recurrence of the linear generator.	The limitation of this work is that they did not assess the device consistency in a mobile IoT. Therefore, they will analyze the performance and reliability of developed frameworks in the transportation system and mobile-based IoT network.

## Data Availability

The data presented in this study are available on request from the corresponding author.

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
