# Peer review of "Applications of Wireless Sensor Networks and Internet of Things Frameworks in the Industry Revolution 4.0: A Systematic Literature Review"

_sensors, 2022, doi:10.3390/s22062087_

Round 1

Reviewer 1 Report

The authors provided a Systematic Literature Review a wide range of information about Industry 4.0, find research gaps, and recommends future directions. This study mainly focuses on research solutions and new techniques to automate industry 4.0. They analyzed 100+ articles from 2014 till 2021. This paper covers several aspects of Industry 4.0, from the designing phase to security needs, from deployment stage to classification of the network, the difficulties, challenges, and future directions. Below are minor suggestions to improve the paper. 1. The abstract contains a lot of background and too less of the proposed work. Author should add more contributions in the abstract. 2. In abstract, “wireless sensor networks (WSN)” should be written as WSNs. 3. In Keyword list, authors should add keywords like SLR, Survey, State-of-the-art 4. The caption of all figures must be descriptive and extensively If you see in figure 3 and other only industry 4.0 is written which is vague, and authors should follow the same color combinations in image. Why authors used green color when there is other color too in figure 3. 5. Figure 4 need be redrawn in new style. 6. Author should improve the conclusion too and recent references in the paper. 7. Enhance the future directions by adding how the research can be carried forward in upcoming Industry 5.0 era. 8. More context about IIoT can be added through out the article.

Author Response

Reviewer 1

Reviewer#1, Concern #1: The abstract contains a lot of background and too less of the proposed work. Author should add more contributions in the abstract.

Author response: Many thanks for noticing this issue. We can confirm that we have addressed this comment.

Author action: We have updated the abstract section as suggested (Section: Abstract, Page:1).

Reviewer#1, Concern #2: In abstract, “wireless sensor networks (WSN)” should be written as WSNs.

Author response: Many thanks for noticing this issue. We can confirm that we have addressed this comment.

Author action: We have removed this mistake and changed as suggested. Changes made in yellow color. (Section: Abstract, Page:1, L: 04)

Reviewer#1, Concern #3: In Keyword list, authors should add keywords like SLR, Survey, State-of-the-art

Author response: Thank you. We can confirm that this comment has now been addressed.

Author action: We have added these new keywords in keyword list highlighted with blue color. (Page:1, L:18).

Reviewer#1, Concern #4: The caption of all figures must be descriptive and extensively If you see in figure 3 and other only industry 4.0 is written which is vague, and authors should follow the same color combinations in image. Why authors used green color when there is other color too in figure 3.

Author response: Many thanks for pointing out this. We can confirm that we have addressed this comment.

Author action: We have made changes as suggested. All the figure captions are self-descriptive. and follow the same color theme for all figures. redraw the figure 3 in blue theme. All changes made, are highlighted in blue color. . (Section: Introductiont, Page:4)

Reviewer#1, Concern #5:  Figure 4 need be redrawn in new style.

Author response: Many thanks for pointing out this. We can confirm that we have addressed this comment.

Author action: We have redrawn figure 4 in new style as suggested . (Section: Research Methodology, Page: 8)

Reviewer#1, Concern #6: Author should improve the conclusion too and recent references in the paper.

Author response: Many thanks for noticing this issue. We can confirm that we have addressed this comment.

Author action: We have rewritten the conclusion and improve recent reference in paper as suggested. Changes are highlighted in blue color. (Section: conclusion, Page:29). New references at the end of paper.

Reviewer#1, Concern #7: Enhance the future directions by adding how the research can be carried forward in upcoming Industry 5.0 era.

Author response: Many thanks for noticing this issue. We can confirm that we have addressed this comment.

Author action: We have modified the future directions as suggested. . (Section: Future directions, Page:29, L-576-581).

Reviewer#1, Concern#8: More context about IIoT can be added through out the article.

Author response: Many thanks for noticing this issue. We can confirm that we have addressed this comment.

Author action: We have added more about IIoT as suggested. Changes made are highlighted in blue color.  (Subsection : RQ2: Contributions of IoT in IR4.0, Page 17-18, L-310-3117, L:319-323).

Reviewer 2 Report

The title of the document suggests systematic literature focused on applications of WSN and IoT in the context of Industry 4.0. This is indeed presented, but the study performed included an additional interest of the authors, that as I read the document had a feeling that it could have been performed separately, and this is the part related to security attacks and network intrusions. Going back to reading the title confirmed this.

While there is a lot of interesting and valuable information, I suggest narrowing the focus to what is looked for in research questions 1, 2, probably 3, and either 6 or 7. Those are more interrelated and will definitely result in an easier to read and even more interesting document for those searching for this type of information.

I have some concerns with the methodology.

  • Table 4 lists the search strings (wrongly labeled as "Research strings") but there is nothing about security and threats. In addition, the study states to find the same type of information for both WSN and IoT, but search string are not designed for this; i.e. "application of WSN" vs "implementation of IoT infrastructure" will not return the same type of results.
  • The title reads "Systematic literature analysis", but the study is also referred as "systematic literature review" and "systematic literature survey". There is a difference between a review and a survey.
  • Inclusion and exclusion criteria are not clear. It seems that the filters for exclusion were only language and date of publication, which should have been applied since the search stage and not be listed as exclusion criteria.

When reporting results, there are several types of network attacks listed and described, but there is not enough information provided on their impact on WSN, IoT, or IR4.0.

Section 4.7 is titled "Limitations of the literature review", and is linked to RQ7, which would explain in a better way the contents of the section. Contributions and limitations of the primary studies are listed on Table 8, but there is no indication or discussion of how this relates to the objectives of this SLR.

Sections 5 and 6 are interesting and enjoyable to read. I recommend including more references, especially in the first part of section 6.

The presentation of the document should be improved:

  • Some tables and images are several pages from the place they are first mentioned.
  • Some tables have caption on the bottom instead of top.
  • Format of references is not the same for all.
  • Please check for grammar issues across all the document.

Author Response

Reviewer 2

Reviewer#2, Concern #1: The title of the document suggests systematic literature focused on applications of WSN and IoT in the context of Industry 4.0. This is indeed presented, but the study performed included an additional interest of the authors, that as I read the document had a feeling that it could have been performed separately, and this is the part related to security attacks and network intrusions. Going back to reading the title confirmed this.

Author response: Many thanks for noticing this problem. We can confirm that we have addressed this comment.

Author action: Security attacks is the 5th Research question of our proposed SLR. check RQ 5 in table 3. highlighted in blue color. (section: Research goals, page :9) It’s not to a separate section.

Reviewer#2, Concern #2: While there is a lot of interesting and valuable information, I suggest narrowing the focus to what is looked for in research questions 1, 2, probably 3, and either 6 or 7. Those are more interrelated and will definitely result in an easier to read and even more interesting document for those searching for this type of information.

Author response: Many thanks for noticing this problem. We can confirm that we have addressed this comment.

Author action: We have modified these sections (RQ1, RQ2 and RQ3) and try to be concise. Irrelevant is removed. also added more references in section. text is highlighted in blue color.

Reviewer#2, Concern #3: Table 4 lists the search strings (wrongly labeled as "Research strings") but there is nothing about security and threats. In addition, the study states to find the same type of information for both WSN and IoT, but search string are not designed for this, i.e. "application of WSN" vs "implementation of IoT infrastructure" will not return the same type of results.

Author response: Many thanks for noticing this problem. We can confirm that we have addressed this comment.

Author action: We have changed this typo mistake and designed a new search string related to attacks and make it part of group 4, which is highlighted in blue color (Table 4) (section:Inclusion/exclusion criteria, page 10)

Reviewer#2, Concern #4: The title reads "Systematic literature analysis", but the study is also referred as "systematic literature review" and "systematic literature survey". There is a difference between a review and a survey.

Author response: Many thanks for noticing this problem. We can confirm that we have addressed this comment.

Author action: This is systematic literature review (SLR). all mistakes are changes throughout the paper as suggested along with title. title changes highlighted in blue.

Reviewer#2, Concern #5: Inclusion and exclusion criteria are not clear. It seems that the filters for exclusion were only language and date of publication, which should have been applied since the search stage and not be listed as exclusion criteria.

Author response: Many thanks for noticing this problem. We can confirm that we have addressed this comment.

Author action: We have added Table 5 to elaborate this point. highlighted in blue color. (section:Inclusion and exclusion criteria, page 10)

Reviewer#2, Concern #6: When reporting results, there are several types of network attacks listed and described, but there is not enough information provided on their impact on WSN, IoT, or IR4.0.

Author response: Many thanks for noticing this problem. We can confirm that we have addressed this comment.

Author action: We have provided the impact of each attack on network performance. Changes are heighted on (section 4.5 : RQ5 and its subsection, page 20) at these lines (L: 419-422 ,L:429-431 L:438-440; L:445-446 ,L411-412).

Reviewer#2, Concern #7: Section 4.7 is titled "Limitations of the literature review", and is linked to RQ7, which would explain in a better way the contents of the section. Contributions and limitations of the primary studies are listed on Table 8, but there is no indication or discussion of how this relates to the objectives of this SLR.

Author response: Many thanks for noticing this problem. We can confirm that we have addressed this comment.

Author action: We have made changes in limitation section highlighted in blue color (section: Limitations of the literature review, page: 21, L:536-538).

Reviewer#2, Concern #8: Sections 5 and 6 are interesting and enjoyable to read. I recommend including more references, especially in the first part of section 6.

Author response: Many thanks for noticing this problem. We can confirm that we have addressed this comment.

Author action: We have added more future directions. also added new references at the end of paper highlighted in blue. (section: Future directions, page:29, L:576-581).

Reviewer#2, Concern #9: The presentation of the document should be improved: Some tables and images are several pages from the place they are first mentioned.

Author response: Many thanks for noticing this problem. We can confirm that we have addressed this comment.

Author action: We have placed tables and figure on the next page or at possible minimum distance. (Table 2, Figure 4)

Reviewer#2, Concern #10: Some tables have caption on the bottom instead of top.

Author response: Many thanks for noticing this problem. We can confirm that we have addressed this comment.

Author action: we have changes caption from bottom to top (table 7,8)

Reviewer#2, Concern #11: Format of references is not the same for all.

Author response: Many thanks for noticing this problem. We can confirm that we have addressed this comment.

Author action: All references are now in same format.

Reviewer#2, Concern #12: Please check for grammar issues across all the document.

Author response: Many thanks for noticing this problem. We can confirm that we have addressed this comment.

Author action: Many thanks for the suggestion. We can confirm that we have now proofread the paper ourselves and by a native English speaker.

Reviewer 3 Report

Please find my detailed comments in the attached file. In summary, I acknowledge that you have done a lot of work, but your presentation is very unorganized and the benefit for the reader does not become clear at all. I believe that the paper length could be reduced by half if you concentrate on the relevant stuff and instead of only listing descriptions of others' works, summarize more and abstract away from the details.

Author Response

Reviewer 3

Reviewer#3, Concern #1: The first half of the abstract (until “… globally.”) is completely irrelevant for the paper, and even misleading through its concentration on security. It is also wrong in the first sentence – why should there have to be a managing entity. I suggest to remove this first half and rather use the space for more in-depth descriptions of what is in the article.

Author response: Many thanks for mentioning this. We can confirm that we have addressed this comment.

Author action: We have rewritten the abstract part as suggested by reviewer. changes made highlighted in blue color. (Section :Abstract, Page 1)

Reviewer#3, Concern #2:  L-29: “a collection of nodes or many sensor nodes”. What do you want to express with this differentiation? Is this not the same?

Author response: Many thanks for noticing this problem. We can confirm that we have addressed this comment.

Author action: These are same. This mistake is corrected as suggested and highlighted in blue color. (section:Introduction, Page 2, L :26-28)

Reviewer#3, Concern #3:  L 30 “as shown in Figures 1”. No Figures 1 does not show sensor nodes measuring environmental conditions. It shows an architecture of a whole information system. Please be more precise or use a different image.

Author response: Many thanks for noticing this problem. We can confirm that we have addressed this comment.

Author action:  We have mentioned legends in Figure no 1. double round circles are sensor nodes representation. These nodes may be of any type (like temp) sensing from environment. (Please see figure 1)  (section:Introduction, Page 2)

Reviewer#3, Concern #4: L31 transform → transfer

Author response: Many thanks for noticing this problem. We can confirm that we have addressed this comment.

Author action: We have removed this typo mistake as suggested and highlighted in blue color. (section:Introduction, Page 2, L29)

Reviewer#3, Concern #5: L33-35: grammar is wrong, I don’t understand the sentence

Author response: Many thanks for noticing this problem. We can confirm that we have addressed this comment.

Author action:  We have rewritten these sentences as suggested and highlighted in blue color. (section: Introduction, Page 2,  L:29- 33)

Reviewer#3, Concern #6: L36: “primarily”? Can you prove this? I suggest “often”.

Author response: Many thanks for noticing this problem. We can confirm that we have addressed this comment.

Author action:  We have modified as suggested and highlighted in blue color. (section: Introduction, Page 2, L34)

Reviewer#3, Concern #7: L39/40: What is “start-star topology”? Do you have a reference? What does this have to do with ad-hoc network and self-driven?

Author response: Many thanks for noticing this problem. We can confirm that we have addressed this comment.

Author action:  We have modified the whole sentence as suggested and highlighted in blue color.  this was typo mistake. (L-38-39).

Reviewer#3, Concern #8: L40 “Furthermore…” at least make a new paragraph here. This has nothing to do with the sentence before.

Author response: Many thanks for noticing this problem. We can confirm that we have addressed this comment.

Author action: We have modified the sentences and make a new paragraph. changes made are highlighted in blue color. (section: Introduction, Page 2, L38-39).

Reviewer#3, Concern #9: L43/44: repetition, leave this out.

Author response: Many thanks for noticing this problem. We can confirm that we have addressed this comment.

Author action: We have rewritten these few lines in concise way and highlighted in blue color. (section: Introduction, Page 2,L45-47)

Reviewer#3, Concern #10: L45: you start another description of WSN applications. Please rewrite this whole section and make it more concise. The organization is a mess.

Author response: Many thanks for noticing this problem. We can confirm that we have addressed this comment.

Author action: We have rewritten this section in concise way and improve the flow of this section. highlighted in blue at (section: Introduction, Page 2,L:40- 47). also simplified subsection 1.3 (section: paper organization, Page 5, L:101-102).

Reviewer#3, Concern #11: L60: sorry, but also this image does not show “IoT, WSN and its elements”. Please be more precise!

Author response: Many thanks for noticing this problem. We can confirm that we have addressed this comment.

Author action: Figure 3 is an example of IR 4.0. CPS system is the complete description of IR 4.0.  whole integrated process is composed of all these 3 things. We have redrawn the figure 3. Hope now this will be clear.  (section: Introduction, Page 4,Figure 3)

Reviewer#3, Concern #12: L63: statics → statistics

Author response: Many thanks for noticing this problem. We can confirm that we have addressed this comment.

Author action: We have changed in the text as suggested. highlighted in blue color (section: Introduction, Page 3, L: 63)

Reviewer#3, Concern #13: L63/64: “more than 10 billion devices”. You don’t give a reference. Where is this number from?

Author response: Many thanks for noticing this problem. We can confirm that we have addressed this comment.

Author action: We have cited the reference in the text as suggested highlighted in blue color. (section: Introduction, Page 3, L:64)

Reviewer#3, Concern #14: Between L69 and 70: there is a break here. I cannot see the link between the business numbers and the description of what constitutes the survey.

Author response: Many thanks for noticing this problem. We can confirm that we have addressed this comment.

Author action: We have made changes in text suggested and highlighted in blue color. (section: Introduction, Page 3, L69-71)

Reviewer#3, Concern #15: L70-76: it does not become clear what makes the difference between the other papers and yours. Please be very precise here because this is crucial. I you cannot make the difference clear, a reader cannot tell what the purpose of you work is.

Author response: Many thanks for noticing this problem. We can confirm that we have addressed this comment.

Author action: We have made changes in the text as suggested and highlighted in blue color. (section: Introduction, Page 3-4, L75-78)

Reviewer#3, Concern #16: Section 1.1: the motivation remains unclear, it is much too fuzzy. Please concentrate on Industry 4.0 and not on “improving the life of individuals” etc. Say very clearly why you do this SLR! Otherwise, one cannot see the need for it.

Author response: Many thanks for noticing this problem. We can confirm that we have addressed this comment.

Author action: We have rewritten the motivation and now focus is on IR 4.0. Changes highlighted in blue color. (L:80-90).

Reviewer#3, Concern #17: Same with Section 1.2. Too unprecise. What’s the purpose of the first sentences?? They have nothing to do with Industry 4.0. The rest of this subsection is unclear in its structure. Please say very clearly what your paper brings to the reader.

Author response: Many thanks for noticing this problem. We can confirm that we have addressed this comment.

Author action: We have rewritten the subsection 1.2 highlighted in blue color. (section: motivation, Page 4, L:92-97). we have modified 1.3 subsection (section: contribution, 4).

Reviewer#3, Concern #18: Section 2, L108: what do you want to express? Which proposed work?

Author response: Many thanks for noticing this problem. We can confirm that we have addressed this comment.

Author action: We have changed in the text as suggested. proposed work means conducted systematic literature review ((section: related studies, page:6,  L: 107-110).

Reviewer#3, Concern #19: L109: why only some of the related work? Why not all? I think you have to make, at some point, much clearer that you handle two sorts of papers: review articles and original research. Unfortunately, you do not manage to make these differences clear.

Author response: Many thanks for noticing this problem. We can confirm that we have addressed this comment.

Author action: We have made this point clear as suggested and highlighted in blue color (section: related studies, page:6, L:107-110)

Round 2

Reviewer 2 Report

Authors properly addressed several concerns from the previous version. However, there are still some corrections or improvements that should be made:

  • Page 2, line 41. Please explain "Transportation reduces miles into kilometer"
  • Figure 2: Please correct misspellings in the icon for "Environmental Monitoring"
  • Half of the abbreviations in Table 1 are only used once in the document. This table should be reduced and left only those that are used thoroughly.
  • Page 17, Line 308. The use of "cyberattacks" and "cyber-attacks".
  • Page 24, Line 536. "Table ??" where I guess should be "Table 9".

Author Response

Reviewer 2

Reviewer#2, Concern #1: Page 2, line 41. Please explain, "Transportation reduces miles into kilometers."

Author response: Many thanks for noticing this issue. We have addressed this comment.

Author action: We have explained the line as suggested & highlighted it in blue. (Section: Introduction, Page:2, L-41-44).

Reviewer#2, Concern #2: Figure 2: Please correct misspellings in the icon for "Environmental Monitoring."

Author response: Many thanks for noticing this issue. We can confirm that we have addressed this comment.

Author action: We have corrected the spellings in figure 2 as suggested. (Section: Introduction, Page:3, figure 2)

Reviewer#2, Concern #3: Half of the abbreviations in Table 1 are only used once in the document. This table should be reduced and left only those that are used thoroughly.

Author response: Many thanks for noticing this issue. We can confirm that we have addressed this comment.

Author action: We have updated table 1 and removed those used once in the text as suggested (Subsection: Paper organization, Page:5).

Reviewer#2, Concern #4: Page 17, Line 308. The use of "cyberattacks" and "cyber-attacks."

Author response: Many thanks for noticing this issue. We can confirm that we have addressed this comment.

Author action: We have corrected as suggested highlighted in blue color. (Page:17, L:311).

Reviewer#2, Concern #5: Page 24, Line 536. "Table ??" where I guess should be "Table 9".

Author response: Many thanks for noticing this issue. We can confirm that we have addressed this comment.

Author action: We have cited properly as suggested highlighted in blue color. (Section: Limitation and future direction, Page:24, L539).

Reviewer 3 Report

Thanks for making all these changes. I think the paper has improved a lot. I basically have two suggestions:

  • Add, at the appropriate place of your choice, a clear definition of the four main notions of your paper, i.e., WSN, IoT, CPS, and Industry 4.0. It still remains unclear what the differences are.
  • Do extensive editing of the English language - there are still quite a few issues. There is also one "Table ??" reference somewhere in the text. 
  • In the blue text that you added, there is one line that I do not understand at all: Line 41: " Transportation reduces miles into Kilometer at a much low cost " What does that mean?

Author Response

Reviewer 3

Reviewer#3, Concern #1: Add, at the appropriate place of your choice, a clear definition of the four main notions of your paper, i.e., WSN, IoT, CPS, and Industry 4.0. It remains unclear what the differences are.

Author response: Many thanks for noticing this issue. We can confirm that we have addressed this comment.

Author action: We have added these four points as suggested highlighted in blue color (Section: introduction, Page:3, L:64-70).

Reviewer#3, Concern #2: Do extensive editing of the English language - there are still quite a few issues.

Author response: Many thanks for noticing this issue. We can confirm that we have addressed this comment.

Author action: We have resolved issues we found throughout the paper as suggested.

Reviewer#3, Concern #3:  There is also one "Table ??" reference somewhere in the text.

Author response: Many thanks for noticing this issue. We can confirm that we have addressed this comment.

Author action: We have adequately cited as suggested highlighted in blue (Section: Limitation and future direction, Page:24, L539).

Reviewer#3, Concern #4: In the blue text that you added, there is one line that I do not understand at all: Line 41: " Transportation reduces miles into Kilometer at a much low cost " What does that mean?

Author response: Many thanks for noticing this issue. We can confirm that we have addressed this comment.

Author action: We have explained the line as suggested highlighted in blue. (Section: Introduction, Page:2, L-41-44).